# Beyond Row-Level Prediction: A Unified Evaluation of Table Representation Methods and Recoverable Table-Level Geometry

## Abstract

Tables exhibit multiple interacting levels of structure, and useful table representations must compose fine-grained local signals into reusable whole-table embeddings. Yet table representation methods are still often assessed through row-level prediction or downstream supervised tasks rather than through the quality of the table-level representations they produce. We introduce a unified evaluation framework for table representation methods built around four practical desiderata: consistency under partial views, discriminability across label granularities, robustness to benign perturbations, and efficiency. Across controlled synthetic families and real open-source corpora, we find a consistent pattern: lightweight schema- and text-based methods often outperform naive mean-pooled embeddings from state-of-the-art tabular foundation models on the practical quality-cost frontier under this representation-centric evaluation. This suggests that table-level representations are not an automatic byproduct of predictive training, but depend critically on how local tabular signals are composed into a global representation. To test this hypothesis, we freeze the encoder of a tabular foundation model and train lightweight representation heads on top of its outputs. The learned heads substantially improve table-level geometry over naive pooling, showing that useful compositional structure is recoverable from the same encoder states, although bounded by the information ceiling of the frozen backbone. A closed-corpus parent-table retrieval proof of concept mirrors the benchmark trends and again shows simple methods outperforming pooled tabular foundation models. Together, these results position table-level representation as a first-class problem beyond row-level prediction and highlight learned composition as a key ingredient in reusable representations.

## 1 Introduction

Foundation models transformed language and vision not only by improving task accuracy, but also by producing reusable representations that support retrieval, transfer, and lightweight adaptation (Devlin et al., 2019; Radford et al., 2021; Caron et al., 2021). In tabular learning, however, this second promise remains unresolved. Modern tabular foundation models are often strong predictive systems, yet they are rarely evaluated as generators of reusable *table-level* representations. This distinction matters because many practical operations, including indexing, approximate deduplication, overlap estimation, retrieval, filtering, and corpus curation, act on whole tables rather than on individual rows or cells.

Current Tabular Foundation Models (TFMs) predominantly adopt predictive training interfaces, often in the Prior-Data Fitted Network or in-context learning regime (Müller et al., 2022; Hollmann et al., 2023; 2025; Spinaci et al., 2025). These models are optimized for row-level classification or regression under finite context windows, not for preserving a stable geometry over partial views of an entire table. As a result, predictive competence does not automatically imply reusable table-level geometry. A model may be excellent at row-level prediction while still producing weak table representations after naive pooling. Recent tabular work has accordingly focused either on predictive performance, on finer-grained row- and cell-level representations,

or on intrinsic properties of table embeddings (Hoppe et al., 2025; Cong et al., 2023), leaving whole-table representation quality under-specified.

More broadly, a table can be represented in many ways, from lightweight schema and statistical fingerprints to serialization-based text embeddings, specialized table encoders, and pooled representations extracted from tabular foundation models (Chen et al., 2023; Pugnaloni et al., 2025). This raises a more basic question than model comparison alone, namely what makes a good reusable table representation. We argue that this is naturally a compositional question. Tables exhibit multiple interacting levels of structure, and a useful table representation must compose fine-grained local signals, including schema, values, and partial structure, into a reusable whole-table embedding that remains meaningful across views, perturbations, and similarity notions.

In this work, we introduce a unified evaluation framework for table representation methods built around four practical desiderata: consistency under partial views, discriminability across label granularities, robustness to benign perturbations, and efficiency. To support representative evaluation, we sample partitions from source tables and compare diverse representation families under a shared observation interface, including lightweight hashing and statistical baselines, serialization-based text methods, specialized table encoders, and pooled representations from state-of-the-art TFMs such as TabPFN (Hollmann et al., 2023) and CONTEXTTAB (Spinaci et al., 2025). Our evaluation goes beyond downstream accuracy and instead asks whether a representation preserves overlap-sensitive structure under partial views, separates tables under multiple notions of similarity, remains stable under benign perturbations, and does so efficiently. This perspective is also aligned with recent work that treats table overlap as a structural relation in its own right, rather than only as a downstream prediction target (Pugnaloni et al., 2025; Zecchini et al., 2024).

To ensure that the analysis is both controlled and empirically grounded, we evaluate on both synthetic and real open-source corpora. We first construct synthetic families with known generative factors, including physics formulas, geometric shapes, temporal reasoning, and Structural Causal Model priors (Qu et al., 2025). We then evaluate on established real-world benchmarks, including CARTE, OpenML-CC18, and OpenML-CTR23 (Kim et al., 2024; Bischl et al., 2021; Fischer et al., 2023). Across both settings, we find a consistent pattern: lightweight schema- and text-based methods often outperform naive mean-pooled embeddings from state-of-the-art tabular foundation models on the practical quality–cost frontier. This suggests that table-level representation is not an automatic byproduct of predictive training, but depends critically on how local tabular signals are composed into a global representation.

The paper does not stop at diagnosis. We use the benchmark to test whether the observed weakness is intrinsic to the encoder or instead reflects a mismatch between predictive pretraining and table-level aggregation. Concretely, we freeze a pretrained CONTEXTTAB backbone (Spinaci et al., 2025) and train lightweight representation heads on top of its outputs. These learned heads substantially improve table-level geometry over naive pooling, showing that useful compositional structure is recoverable from the same encoder states, although bounded by the information ceiling of the frozen backbone. Finally, we add a closed-corpus parent-table retrieval proof of concept from partial views. The results mirror the benchmark trends, showing both that simple methods continue to outperform pooled tabular foundation models in this source-recovery setting and that learned aggregation over a frozen backbone yields clear gains over native pooling.

**Contributions.**

- We introduce a unified evaluation framework for reusable table-level representations centered on consistency under partial views, discriminability across multiple label granularities, robustness to benign perturbations, and efficiency.
- Across synthetic and real data, we show that lightweight schema- and text-based methods often outperform naive pooled representations from strong predictive tabular foundation models on the practical quality-cost frontier.
- We present a targeted frozen-backbone case study showing that a substantial part of this gap is recoverable by learning lightweight representation heads over encoder outputs, without retraining the encoder itself.
- We provide a closed-corpus parent-table retrieval proof of concept showing that the benchmark trends are consistent with a simple source-recovery setting tied to partial-view consistency and identity-sensitive discrimination.

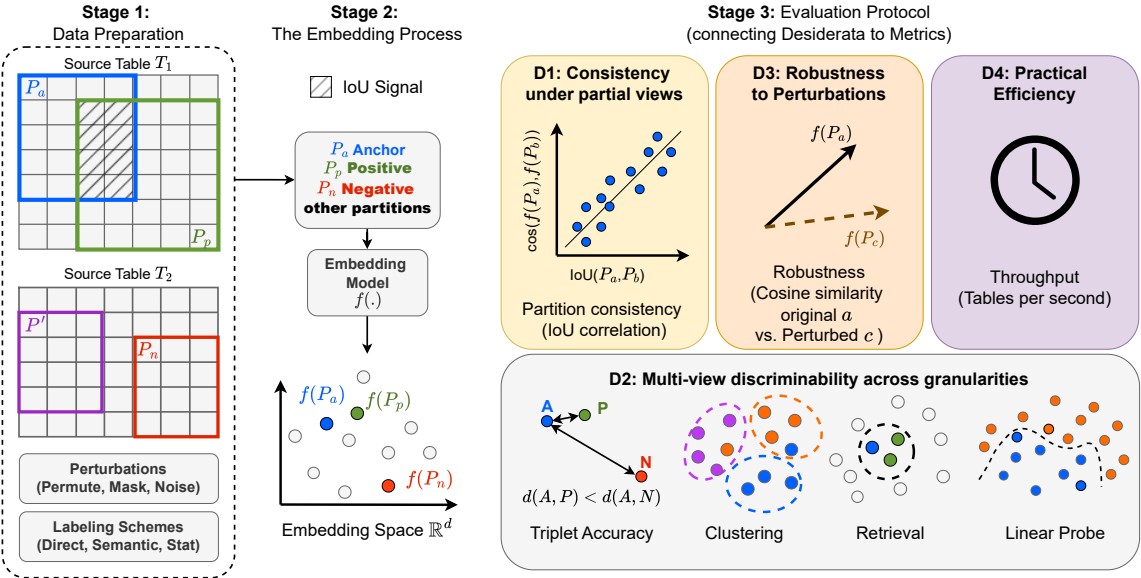

Figure 1: The proposed evaluation pipeline linking table-representation desiderata to concrete evaluation. In stage 1, we sample partitions from source tables and attach labeling schemes and perturbations. Stage 2 embeds each partition into a vector space. Stage 3 measures four desiderata through concrete metrics, allowing comparison of both lightweight baselines and foundation-model-derived representations under the same observation interface.

## 2 Related Work

Tabular foundation models have recently achieved strong predictive performance through prior-data fitted or in-context learning interfaces, including TabPFN and its extensions and newer semantics-aware models such as CONTEXTTAB (Müller et al., 2022; Hollmann et al., 2023; 2025; Spinaci et al., 2025). These models are primarily designed and evaluated for row-level classification or regression, not for preserving a reusable geometry over partial views of whole tables. Our work is motivated by this gap. Rather than asking whether such models predict well, we ask whether their hidden states yield strong *table-level* representations after aggregation.

A second line of work studies tabular or relational representations more directly. Recent efforts have characterized properties of relational-table embeddings and proposed evaluation tools for analyzing them (Cong et al., 2023), while other work compares task-agnostic tabular embeddings from foundation models and classical feature pipelines through downstream learning and anomaly-detection tasks (Hoppe et al., 2025). Our paper is complementary to these directions. We focus on reusable *table-level* representations under a shared observation interface and evaluate them through four practical desiderata: consistency under partial views, discriminability across multiple similarity notions, robustness to benign perturbations, and efficiency.

A third line of work proposes specialized representation methods for structured tables or targets related table-level similarity tasks. In our benchmark, this includes methods such as HyTREL, which encodes tables through hypergraph structure, and Armadillo, which learns graph-based table embeddings for overlap estimation (Chen et al., 2023; Pugnaloni et al., 2025). Relatedly, recent work has formalized largest-table-overlap detection as a problem in its own right (Zecchini et al., 2024). Rather than proposing a single specialized encoder or solving one downstream task directly, we provide a unified benchmark that compares lightweight fingerprints, specialized table encoders, and foundation-model-derived embeddings under the same partial-view interface, and we use that benchmark to test whether stronger table-level geometry can be recovered from a frozen predictive backbone.

Taken together, prior work motivates the need for reusable table representations, but still leaves under-specified how such representations should be evaluated across partial views, similarity granularities, robustness, and efficiency under a shared observation interface. Our benchmark is designed to make that evaluation target explicit.

# 3 Desiderata for Table-level Embeddings

We treat table-level embeddings as reusable interfaces for corpus-scale operations such as retrieval, ranking, and lightweight supervision, without requiring downstream fine-tuning. Due to the lack of existing comparable benchmarks and the scarcity of labeled ground truth data, we design a set of proxy tasks that address this gap and provide a sound evaluation pipeline. The desiderata motivate the proxy task definitions and metrics introduced in Section 4. Figure 1 depicts the connection between the desiderata and the evaluation metrics.

**D1: Consistency under partial views.** In many real-world applications, a table is rarely observed in full; it may be accessed via cached extracts, subsampled partitions, or truncated context windows. A table embedding should therefore be stable across partial views of the same table, with similarity increasing monotonically with view overlap. This property directly supports overlap estimation and approximate deduplication in large corpora, where near-duplicate tables often share only partial content, and enables view-consistent filtering and sampling when curating training data for tabular foundation models.

**D2: Discriminability across label granularities.** A table embedding should support grouping and separation under different label granularities. This is important for training-time curation, where different operations require different equivalence relations, with identity-sensitive signals helping deduplication, semantic groupings support stratified sampling across domains and tasks, and coarse statistical groupings enable type-aware filtering and balancing. The evaluation should characterize both local neighborhood structure (triplet comparisons) and global grouping behavior (clustering), alongside operational proxies such as retrieval performance and linear probe decodability of labels.

**D3: Robustness to benign perturbations.** The table representation should be invariant under row and column permutations, and should remain stable under mild corruption such as masking and small additive noise . Robustness is essential not only for reliable indexing and nearest-neighbor retrieval, but also for deduplication and corpus curation over automatically extracted tables.

**D4: Practical efficiency.** Target applications often involve large corpora; therefore, embedding extraction must be computationally efficient. We measure the throughput (tables per second). An efficient table embedding method balances high throughput with robustness under the aforementioned desiderata, capturing the quality-cost trade-off.

# 4 Benchmark Setup

We study table-level representations through a shared evaluation interface. A table representation method is a function $f$ that maps a table $T$ to a fixed-dimensional vector $f(T) \in \mathbb{R}^d$. We compare two tables using cosine similarity, $\text{sim}(T_a, T_b) = \cos\big(f(T_a), f(T_b)\big)$. Although such methods conceptually operate on full tables, our benchmark applies $f(\cdot)$ to *table partitions* sampled from the same source table. These partitions are valid subtables and allow us to compare diverse representation methods under controlled partial views, perturbations, and shared input budgets.

We evaluate a broad set of table representation methods spanning lightweight fingerprints, serialization-based text encoders, specialized relational encoders, and representations derived from tabular foundation models. Methods that emit row- or cell-level outputs are reduced to a single table vector by a fixed pooling rule, while serialization-based methods operate on a deterministic CSV-style representation of the same table view. To assess discriminability under different notions of similarity, we attach labels at multiple granularities. *Direct* labels correspond to source-table identity, *Semantic* labels capture higher-level grouping, *Semantic+Difficulty* further refines synthetic semantic labels by generator difficulty, and *Stat* labels encode coarse statistical signatures. The main text summarizes the benchmark structure; full method definitions, pooling details, label construction, protocol details, and hyperparameters are deferred to Appendix A.1, A.2, and A.5.

Our proxy tasks are aligned with the desiderata introduced in Section 3. For each source table, we sample multiple partitions under a fixed observation budget and compare their representations through the criteria summarized in Table 1. In particular, D1 evaluates whether representation similarity is consistent with partition overlap, D2 evaluates grouping and separation under multiple notions of table similarity, D3 evaluates invariance or stability under benign perturbations, and D4 measures the practical quality–cost trade-off through throughput.

Table 1: Benchmark summary. Top: families of table representation methods compared under the shared observation interface. Bottom: desiderata, main metrics, and the corresponding evaluation targets. Full details are provided in Appendix A.1 and Appendix A.2.

**(A) Table representation methods**

| Family | Methods | Main signal / representation |
|---|---|---|
| Structural & statistical | HashingSchema, SchemaContent, TableStatistics, StatisticalSummary, MatrixFactorization | Schema tokens, metadata, summary statistics, and numeric spectrum; direct table vectors. |
| Serialization-based text | HashingText, LLMText | CSV-style serialization encoded by feature hashing or a sentence-transformer. |
| Specialized representation | TableVectorizer, HyTREL, Armadillo | Mixed-type row vectorization, hypergraph encoding, or graph neural encoding with pooled table vectors. |
| Foundation-model-derived | TabPFN, CONTEXTTAB | Pretrained row- or cell-level hidden states from tabular foundation models, pooled into a single table vector. |

**(B) Desiderata and metrics**

| Des. | Main metric(s) | What it measures | Evaluation target |
|---|---|---|---|
| D1 | Spearman correlation between IoU and cosine similarity | Whether representation similarity tracks overlap across partial views of the same source table | Partial-view consistency |
| D2 | TR-Avg, CL-Avg, Recall@5, linear probe accuracy | Whether representations support local ranking, global grouping, retrieval, and label decoding under multiple labeling schemes | Discriminability across *Direct*, *Semantic*, *Semantic+Difficulty* (synthetic only), and *Stat* labels |
| D3 | Cosine similarity after permutation, masking, and typed noise | Whether representations remain stable under benign perturbations | Robustness |
| D4 | Tables per second | Embedding throughput under the shared observation interface | Efficiency |

## 5 Main Benchmark Results: Revealing a Representation-Prediction Tension

We evaluate on both synthetic and real open-source corpora in order to balance controlled structure with real-world variability. Across both settings, we sample 100 source tables and compute the benchmark metrics on $K_p = 10$ partitions per table under a shared observation budget. We repeat the full evaluation over 10 random seeds, yielding 10,000 sampled partition instances per embedder before metric-specific pair, triplet, and retrieval constructions. The synthetic benchmark includes families with controlled generative factors spanning formulaic, geometric, temporal, and SCM-based structure, while the real-data benchmark is built from CARTE, OpenML-CC18, and OpenML-CTR23. Full dataset construction details, sampling configurations, and label definitions are deferred to Appendix A.4 and Appendix A.5. We use the benchmark results below to analyze the behavior of different table representation methods under a shared evaluation interface and to establish the main empirical pattern of the paper, namely that reusable table-level representation quality does not automatically follow row-level predictive strength. This pattern in turn motivates the frozen-backbone case study and the downstream source-recovery proof of concept.

### 5.1 Synthetic benchmark results

On synthetic data (Table 2, Figure 2), lightweight hashing and statistical-summary embedders remain among the strongest methods, with HashingSchema and HashingText leading the updated quality-only ranking. HashingSchema achieves the best D1 consistency and the best overall average rank among the baseline embedders, while HashingText provides the strongest D2 retrieval and linear probe signals (both near ceiling on Recall@5 and LP). LLMText yields the strongest D2 clustering alignment (Table 2). Taken together, these results show that lightweight methods dominate the table-level quality frontier on synthetic data,

Table 2: Evaluation Results for Synthetic data (mean ± 95% CI over 10 seeds).

| Embedder | D1 Consistency | | D2 Label-Based - Avg | | | | | D3 Robustness | | | | Overall |
|---|---|---|---|---|---|---|---|---|---|---|---|---|
| | Spearman | Rank | TR-Avg | CL-Avg | R@5 | LP | Rank | Perm | Noise | Mask | Rank | Avg Rank |
| HashingSchema | **0.58**±0.03 | 1.00 | **0.54**±0.01 | 0.54±0.02 | 0.72±0.02 | 0.61±0.01 | 4.50 | 1.00±0.00 | 0.98±0.00 | 0.99±0.00 | 5.17 | 3.56 |
| SchemaContent | 0.28±0.03 | 9.00 | 0.01±0.00 | 0.25±0.01 | 0.78±0.01 | 0.59±0.01 | 8.75 | 1.00±0.00 | 1.00±0.00 | 1.00±0.00 | 3.33 | 7.03 |
| TableStatistics | 0.34±0.03 | 6.00 | 0.14±0.01 | 0.35±0.01 | 0.71±0.01 | 0.51±0.02 | 9.25 | 1.00±0.00 | 1.00±0.00 | 1.00±0.00 | 3.50 | 6.25 |
| StatisticalSummary | 0.39±0.05 | 4.00 | 0.36±0.01 | 0.45±0.02 | 0.90±0.01 | 0.56±0.02 | 4.50 | 1.00±0.00 | 0.99±0.00 | 1.00±0.00 | 4.17 | 4.22 |
| MatrixFactorization | 0.37±0.04 | 5.00 | 0.22±0.01 | 0.32±0.01 | 0.78±0.03 | 0.50±0.02 | 8.00 | 1.00±0.00 | 1.00±0.00 | 1.00±0.00 | 2.17 | 5.06 |
| HashingText | 0.54±0.05 | 2.00 | 0.47±0.01 | 0.46±0.03 | **0.99**±0.00 | **0.99**±0.01 | 2.00 | 1.00±0.00 | 0.66±0.02 | 0.97±0.00 | 7.50 | 3.83 |
| LLMText | 0.30±0.02 | 8.00 | 0.51±0.01 | **0.55**±0.03 | 0.87±0.01 | 0.77±0.02 | 2.25 | 0.97±0.00 | 0.98±0.00 | 0.99±0.00 | 7.00 | 5.75 |
| TableVectorizer | 0.32±0.02 | 7.00 | 0.18±0.01 | 0.21±0.01 | 0.74±0.01 | 0.40±0.02 | 10.00 | 0.31±0.01 | 1.00±0.00 | 0.82±0.02 | 9.00 | 8.67 |
| HyTREL | 0.18±0.04 | 10.00 | 0.25±0.01 | 0.33±0.02 | 0.83±0.01 | 0.74±0.01 | 5.00 | 0.99±0.00 | 0.98±0.00 | 0.89±0.00 | 8.67 | 7.89 |
| Armadillo | 0.41±0.02 | 3.00 | 0.31±0.01 | 0.30±0.02 | 0.88±0.01 | 0.73±0.02 | 5.00 | 1.00±0.00 | 0.47±0.01 | 0.87±0.00 | 8.83 | 5.61 |
| TabPFN | -0.13±0.03 | 12.00 | 0.16±0.01 | 0.25±0.02 | 0.72±0.01 | 0.51±0.03 | 9.50 | 0.76±0.01 | 0.94±0.01 | 0.99±0.00 | 9.33 | 10.28 |
| CONTEXTTAB | 0.06±0.02 | 11.00 | 0.18±0.00 | 0.27±0.03 | 0.70±0.01 | 0.57±0.02 | 9.25 | 0.97±0.00 | 0.97±0.00 | 0.97±0.00 | 9.33 | 9.86 |

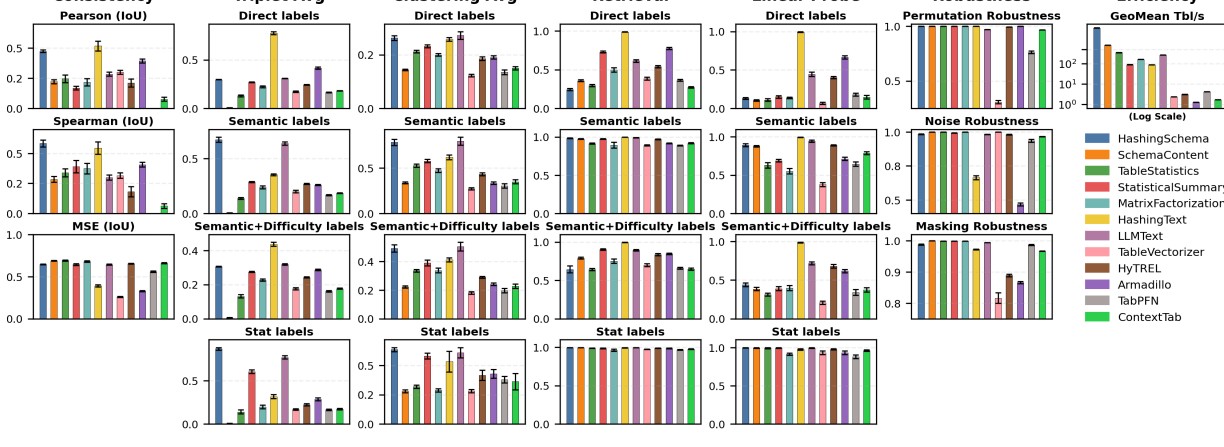

Figure 2: Metric summary for Synthetic data across the benchmark dimensions.

whereas pooled prediction-oriented TFM states remain weak on the criteria most closely tied to reusable table representations.

D2 metrics are label-dependent (Appendix A.6). The appendix confirms this specialization in more detail. For *Direct* identity labels, HashingText achieves the strongest triplet ranking under hard and conditional-hard negatives (Table 11 in Appendix). For the *Stat* labeling scheme, HashingSchema is near-ceiling on both triplet and clustering metrics (Table 14 in Appendix). Increasing label granularity from *Semantic* to *Semantic+Difficulty* reduces hard-negative triplet accuracy across most schema-centric and statistical summary methods (Table 13 in Appendix), while text/content-based methods remain comparatively stronger on strict triplet ranking.

For D3 robustness, most methods saturate on permutation robustness, with TableVectorizer as the clear exception due to its ordering sensitivity (Table 2). Noise robustness is most challenging for value-/content-sensitive methods (Figure 2). This makes the main failure mode of pooled TFMs more specific. They are not uniformly weak across all metrics, but remain consistently weak on D1 and D2, where reusable table-level geometry matters most. Controlled D4 results in Appendix A.7 further strengthen this picture, since lightweight fingerprinting methods are orders of magnitude faster, while learned heads on frozen CONTEXTTAB recover quality without materially changing the underlying throughput.

## 5.2 Real-data benchmark results

On real open-source data (Table 3, Figure 3), lightweight hashing and statistical-summary embedding models remain consistently among the top-performing methods. HashingText ranks best overall, combining the strongest D1 consistency with strong D2 retrieval and linear probe performance, while HashingSchema follows closely, reflecting the continued importance of schema/type cues in heterogeneous benchmark suites.

Table 3: Evaluation Results for Real data (mean ± 95% CI over 10 seeds).

| Embedder | D1 Consistency | | D2 Label-Based - Avg | | | | | D3 Robustness | | | | Overall |
|---|---|---|---|---|---|---|---|---|---|---|---|---|
| | Spearman | Rank | TR-Avg | CL-Avg | R@5 | LP | Rank | Perm | Noise | Mask | Rank | Avg Rank |
| HashingSchema | 0.32±0.05 | 2.00 | **0.56±0.02** | 0.35±0.04 | 0.97±0.01 | 0.87±0.02 | 2.50 | **1.00±0.00** | 0.94±0.01 | 0.95±0.01 | 7.50 | 4.00 |
| SchemaContent | 0.26±0.02 | 3.00 | 0.11±0.01 | 0.17±0.01 | 0.80±0.01 | 0.88±0.02 | 8.00 | **1.00±0.00** | **1.00±0.00** | **1.00±0.00** | 3.00 | 4.67 |
| TableStatistics | 0.22±0.03 | 4.00 | 0.07±0.00 | 0.23±0.01 | 0.77±0.02 | 0.59±0.02 | 8.25 | **1.00±0.00** | **1.00±0.00** | **1.00±0.00** | 2.83 | 5.03 |
| StatisticalSummary | 0.20±0.03 | 6.00 | 0.30±0.01 | 0.30±0.02 | 0.85±0.02 | 0.53±0.02 | 5.75 | **1.00±0.00** | **1.00±0.00** | **1.00±0.00** | 3.50 | 5.08 |
| MatrixFactorization | 0.17±0.03 | 8.00 | 0.21±0.01 | 0.19±0.01 | 0.65±0.02 | 0.28±0.02 | 9.75 | **1.00±0.00** | **1.00±0.00** | **1.00±0.00** | 3.17 | 6.97 |
| HashingText | **0.40±0.04** | 1.00 | 0.51±0.01 | 0.28±0.02 | **0.97±0.01** | **0.89±0.02** | 2.50 | **1.00±0.00** | 0.75±0.01 | 0.98±0.00 | 6.83 | 3.44 |
| LLMText | 0.14±0.04 | 10.00 | **0.56±0.02** | **0.38±0.03** | **0.98±0.00** | 0.88±0.02 | 1.50 | 0.92±0.01 | 0.94±0.00 | 0.99±0.00 | 7.67 | 6.39 |
| TableVectorizer | 0.02±0.03 | 12.00 | 0.18±0.01 | 0.17±0.01 | 0.58±0.01 | 0.30±0.03 | 11.00 | 0.33±0.02 | 0.98±0.00 | 0.84±0.01 | 9.33 | 10.78 |
| HyTREL | 0.16±0.03 | 9.00 | 0.24±0.01 | 0.21±0.01 | 0.86±0.01 | 0.77±0.02 | 5.50 | **0.99±0.00** | 0.89±0.01 | 0.96±0.00 | 9.00 | 7.83 |
| Armadillo | 0.21±0.04 | 5.00 | 0.36±0.01 | 0.22±0.02 | 0.86±0.01 | 0.69±0.03 | 5.50 | **1.00±0.00** | 0.59±0.02 | 0.77±0.01 | 9.17 | 6.56 |
| TabPFN | 0.05±0.04 | 11.00 | 0.13±0.01 | 0.19±0.02 | 0.60±0.01 | 0.42±0.02 | 10.25 | 0.78±0.01 | 0.97±0.01 | 0.97±0.01 | 8.00 | 9.75 |
| CONTEXTTAB | 0.18±0.01 | 7.00 | 0.24±0.01 | 0.21±0.01 | 0.76±0.02 | 0.70±0.02 | 7.50 | 0.97±0.00 | 0.97±0.00 | 0.97±0.00 | 8.00 | 7.50 |

Figure 3: Metric summary for Real data across the benchmark dimensions.

LLMText achieves strong D2 clustering and retrieval signals but ranks poorly on D1 (Table 3). The same qualitative pattern therefore carries over to real data, where simple schema- and text-based methods remain highly competitive or dominant, whereas pooled TFM states lag on the core table-level criteria.

The per-label D2 breakdown (Appendix A.6) shows that the *Direct* labeling scheme is substantially harder on real tables than on synthetic data: hard-negative triplet scores are low across most methods (Table 16 in Appendix), even when retrieval and linear probe remain high. *Stat* labels are the easiest and align strongly with schema cues (Table 18 in Appendix), while *Semantic* labels score in between identity and coarse type groupings (Table 17 in Appendix). Qualitatively, the t-SNE projections (Figure 6 in Appendix) mirror this trend: *Stat* labels form visibly separated regions for schema-driven methods, while *Direct* labels exhibit substantial mixing for most embedding models.

As in synthetic data, D3 permutation robustness is near-saturated for most methods except TableVectorizer (Table 3). Noise robustness again differentiates value-/content-sensitive methods (Figure 3). Finally, pooled representations from TFMs remain weak as table embeddings under D1/D2, with ConTextTab outperforming TabPFN. Together with the synthetic results, this motivates the next question of the paper, namely whether the weakness of pooled TFMs reflects an intrinsic limitation of the encoder or instead a mismatch between predictive pretraining and the rule used to aggregate local encoder states into a table-level representation.

# 6 Case Study: Recovering Table-level Representation from a Frozen Foundation Model

The benchmark above is diagnostic, but it also suggests a concrete hypothesis. The weakness of pooled TFMs may reflect not only backbone limitations, but also a mismatch between prediction-oriented pretraining and the rule used to compose local encoder states into a table-level representation. This is naturally a compositional question. If the frozen encoder already contains useful local structure, then a better aggregation

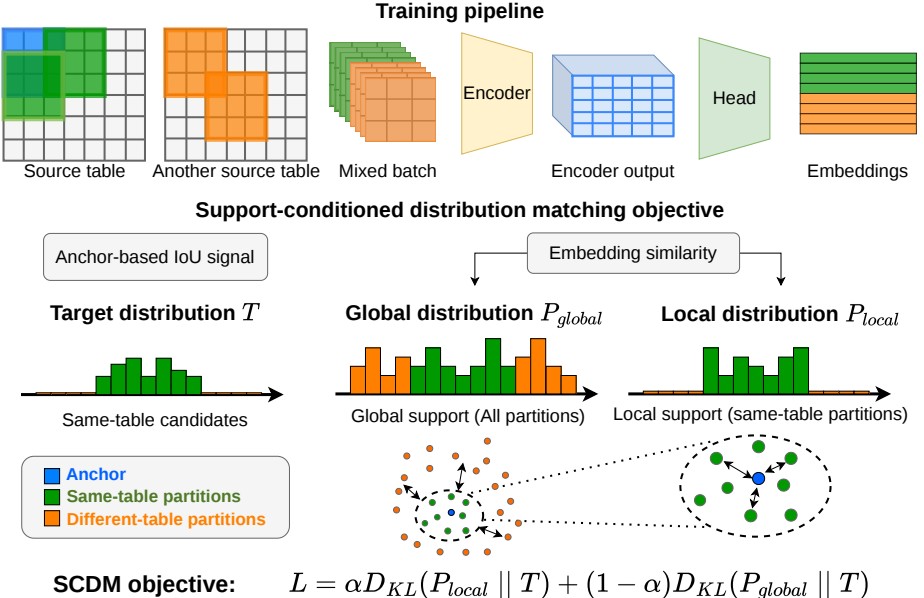

Figure 4: Case-study overview for learning table-level embeddings on top of frozen CONTEXTTAB. At the top, partitions sampled from one source table and from other source tables are collated into mixed batches, encoded by the frozen backbone, and aggregated by a trainable representation head into table embeddings. At the bottom, support-conditioned distribution matching defines a target distribution $T$ from anchor-based overlap and matches it with local and global similarity distributions induced by the learned embeddings.

mechanism should be able to recover stronger table-level geometry from the same states. To test this, we freeze a pretrained CONTEXTTAB backbone and train lightweight representation heads on top of its contextual cell states.

We choose CONTEXTTAB for two reasons. First, it is a semantics-aware tabular foundation model that explicitly processes column titles, which is relevant given the strong performance of schema-sensitive baselines in Section 5. This makes CONTEXTTAB a natural case-study backbone under the present evaluation regime. Second, freezing the pretrained backbone isolates the role of aggregation and auxiliary supervision. Any gains can therefore be attributed primarily to improved table-level composition rather than to additional encoder capacity. We accordingly scope the recoverability claim in this section to CONTEXTTAB; extending the intervention to training from scratch or to other foundation models is left for future work.

For each sampled partition $P$, the frozen encoder produces contextual cell states $H(P) \in \mathbb{R}^{n \times m \times d}$. A trainable representation head then maps $H(P)$ to a single table vector $z(P) \in \mathbb{R}^d$. We study three head families, **Attention**, **GatedAttention**, and **GatedMLP**. Each head is trained on partial views sampled with the same partition interface used throughout the benchmark, enabling direct comparison to the pooled CONTEXTTAB baseline.

Figure 4 summarizes the setup. Same-sized partitions sampled from one source table and from another source table are collated into mixed batches, passed through the frozen backbone, and aggregated by a trainable head into table embeddings. For an anchor partition, we derive an overlap-based target distribution $T$ from anchor-based intersection-over-union over same-table candidates. We then compute two similarity-based distributions from the learned embeddings, a global-support distribution $P_{\text{global}}$ normalized over the full mixed batch and a local-support distribution $P_{\text{local}}$ normalized over same-table candidates only. The resulting objective is

$$\mathcal{L} = \alpha \, D_{\text{KL}}(P_{\text{local}} \,\|\, T) + (1 - \alpha) \, D_{\text{KL}}(P_{\text{global}} \,\|\, T),$$

where $\alpha$ controls the trade-off between fine within-table overlap structure and broader cross-table discrimination. In effect, the head learns how to compose local contextual states into a reusable table representation

Table 4: Representative learned-head configurations on top of frozen CONTEXTTAB. For each dataset and head family, we report the strongest observed configuration under the lowest overall average-rank criterion. Full $\alpha$-sweeps are reported in the Appendix A.10, and comparison to other baselines in Appendix A.8.

| Data | Embedder | $\alpha$ | D1 Consistency | D2 Label-Based - Avg | | | | D3 Robustness | | |
|---|---|---|---|---|---|---|---|---|---|---|
| | | | Spearman | TR-Avg | CL-Avg | R@5 | LP | Perm | Noise | Mask |
| Synthetic | CONTEXTTAB (pooled) | – | 0.06±0.02 | 0.18±0.00 | 0.27±0.03 | 0.70±0.01 | 0.57±0.02 | 0.97±0.00 | 0.97±0.00 | 0.97±0.00 |
| | Attention | 0.75 | **0.50±0.02** | **0.34±0.01** | **0.46±0.02** | 0.88±0.01 | 0.78±0.02 | **1.00±0.00** | **1.00±0.00** | 0.99±0.00 |
| | GatedAttention | 1.00 | 0.48±0.02 | 0.31±0.01 | 0.46±0.03 | **0.89±0.01** | **0.80±0.02** | **1.00±0.00** | **1.00±0.00** | 0.99±0.00 |
| | GatedMLP | 0.75 | 0.48±0.02 | 0.30±0.01 | 0.43±0.04 | 0.88±0.01 | **0.80±0.02** | **1.00±0.00** | **1.00±0.00** | 0.99±0.00 |
| Real | CONTEXTTAB (pooled) | – | 0.18±0.01 | 0.24±0.01 | 0.21±0.01 | 0.76±0.02 | 0.70±0.02 | 0.97±0.00 | 0.97±0.00 | 0.97±0.00 |
| | Attention | 0.25 | **0.27±0.03** | **0.41±0.02** | **0.30±0.03** | **0.96±0.01** | 0.87±0.02 | **1.00±0.00** | 0.99±0.00 | 0.98±0.00 |
| | GatedAttention | 0.50 | 0.26±0.03 | 0.38±0.02 | 0.27±0.02 | 0.95±0.01 | 0.87±0.02 | **1.00±0.00** | 0.99±0.00 | 0.99±0.00 |
| | GatedMLP | 1.00 | 0.24±0.03 | 0.31±0.02 | 0.27±0.03 | 0.95±0.01 | **0.88±0.02** | **1.00±0.00** | **1.00±0.00** | 0.99±0.00 |

while the backbone remains fixed. The training signal is different to the D1 Spearman metric reported in the benchmark, and the loss induces only an implicit contrastive or separation behavior through the local/global batchwise normalization and relies only on Direct same-table identity structure. Additional training details are provided in Appendix A.9.

For readability, the main paper reports one representative configuration per head family and per data regime. Within each regime, we select the strongest observed configuration according to the lowest overall average rank across the D1–D3 quality metrics. The full $\alpha$-sweep is deferred to Appendix A.10, where we show that the main conclusions do not depend on a single fragile hyperparameter choice.

Table 4 shows that the conclusion is robust to this compact reporting. Even with the backbone frozen, all learned heads improve substantially over the original pooled CONTEXTTAB embedding on D1 and D2. On synthetic data, the gains are large. The pooled baseline reaches only 0.06 Spearman on D1, whereas the representative learned heads improve this to 0.48–0.50. At the same time, triplet ranking rises from 0.18 to 0.30–0.34, clustering from 0.27 to 0.43–0.46, retrieval from 0.70 to 0.88–0.89, and linear probing from 0.57 to 0.78–0.80. On real open-source data, the margin is smaller but still systematic. D1 improves from 0.18 to 0.24–0.27, while triplet ranking rises from 0.24 to 0.31–0.41, clustering from 0.21 to 0.27–0.30, retrieval from 0.76 to 0.95–0.96, and linear probing from 0.70 to 0.87–0.88.

Two conclusions follow. First, the weakness of pooled CONTEXTTAB is not solely a backbone limitation. A substantial part of the gap comes from the mismatch between prediction-oriented pretraining and table-level composition. Once the aggregation rule is learned explicitly and aligned with D1/D2, much better table-level geometry becomes recoverable from the same frozen encoder states. Second, the gains are clearly bounded. Because the encoder is frozen, the learned heads can only reorganize information already exposed in the pretrained backbone states. They improve substantially over naive pooling, but they do not yet close the full gap to the strongest lightweight baselines reported earlier. The controlled D4 results in Appendix A.7 further show that these quality gains do not materially change the underlying efficiency regime.

The appendix strengthens this interpretation in two additional ways. First, when the representative learned heads are inserted back into the full benchmark leaderboard and the ranks are recomputed, learned pooling moves CONTEXTTAB from near the bottom of the quality-only ranking to the top on synthetic data and into the top group on real data, without consistently displacing the strongest lightweight baselines (Appendix A.8). Second, the full $\alpha$-sweep shows that the gains are not tied to a single fragile hyperparameter choice. All three head families improve over pooled CONTEXTTAB on D1 and D2 across a broad range of $\alpha$ values, although the best trade-off differs across metrics: D1 often prefers a stronger local component, while D2 benefits from a mixture of local and global supervision (Appendix A.10). Together, these appendix results support the compositional interpretation of the case study. They suggest that the frozen backbone already contains useful local structure, but that turning this structure into a strong reusable table-level representation depends critically on learning the rule that composes local states into a whole-table embedding.

We therefore view this experiment as a diagnostic lower-bound intervention rather than a final answer. It shows that useful compositional structure is already present in CONTEXTTAB and can be recovered by learning a better table-level composition rule, but it also shows that the quality of the final representation remains bounded by the information available in the frozen backbone. In that sense, the case study supports

Table 5: Parent-table retrieval proof of concept on synthetic and real data (mean ± 95% CI over seeds).

| Embedder | Synthetic | | | | Real | | | |
|---|---|---|---|---|---|---|---|---|
| | R@1 | R@5 | R@10 | MRR | R@1 | R@5 | R@10 | MRR |
| HashingSchema | 0.10±0.00 | 0.47±0.01 | 0.78±0.02 | 0.28±0.00 | **0.67±0.02** | **0.97±0.01** | 0.98±0.00 | **0.80±0.01** |
| SchemaContent | 0.01±0.00 | 0.05±0.00 | 0.10±0.00 | 0.05±0.00 | 0.10±0.01 | 0.28±0.02 | 0.38±0.02 | 0.20±0.01 |
| TableStatistics | 0.04±0.00 | 0.19±0.01 | 0.33±0.01 | 0.13±0.00 | 0.04±0.01 | 0.16±0.01 | 0.25±0.02 | 0.11±0.01 |
| StatisticalSummary | 0.32±0.02 | 0.69±0.02 | 0.82±0.02 | 0.48±0.02 | 0.25±0.02 | 0.51±0.03 | 0.63±0.03 | 0.38±0.02 |
| MatrixFactorization | 0.19±0.01 | 0.49±0.03 | 0.61±0.04 | 0.33±0.02 | 0.10±0.01 | 0.27±0.02 | 0.38±0.03 | 0.19±0.01 |
| HashingText | **0.98±0.01** | **0.99±0.00** | **1.00±0.00** | **0.99±0.00** | 0.66±0.02 | **0.97±0.01** | 0.98±0.01 | **0.80±0.01** |
| LLMText | 0.24±0.01 | 0.64±0.02 | 0.85±0.02 | 0.42±0.01 | 0.61±0.03 | **0.97±0.01** | **0.99±0.00** | 0.76±0.02 |
| TableVectorizer | 0.06±0.01 | 0.16±0.01 | 0.22±0.02 | 0.13±0.01 | 0.03±0.01 | 0.12±0.01 | 0.19±0.01 | 0.09±0.01 |
| HyTREL | 0.15±0.01 | 0.40±0.01 | 0.57±0.02 | 0.28±0.01 | 0.29±0.01 | 0.62±0.02 | 0.73±0.02 | 0.44±0.01 |
| Armadillo | 0.63±0.03 | 0.81±0.02 | 0.87±0.02 | 0.71±0.02 | 0.45±0.01 | 0.77±0.01 | 0.84±0.01 | 0.60±0.01 |
| TabPFN | 0.04±0.00 | 0.13±0.00 | 0.21±0.01 | 0.10±0.00 | 0.03±0.01 | 0.11±0.01 | 0.19±0.01 | 0.09±0.01 |
| ConTextTab | 0.05±0.00 | 0.21±0.01 | 0.35±0.01 | 0.14±0.01 | 0.10±0.01 | 0.27±0.02 | 0.39±0.02 | 0.19±0.01 |
| *Learned heads on frozen ConTextTab* | | | | | | | | |
| ConTextTab-R Attention | 0.20±0.01 | 0.50±0.02 | 0.65±0.02 | 0.34±0.01 | 0.35±0.01 | 0.70±0.03 | 0.81±0.02 | 0.51±0.01 |
| ConTextTab-R GatedAttention | 0.16±0.01 | 0.45±0.01 | 0.63±0.01 | 0.30±0.01 | 0.35±0.01 | 0.74±0.02 | 0.86±0.02 | 0.52±0.01 |
| ConTextTab-R Gated MLP | 0.13±0.01 | 0.39±0.01 | 0.55±0.02 | 0.26±0.01 | 0.29±0.01 | 0.63±0.02 | 0.75±0.03 | 0.45±0.01 |

both sides of the paper's main claim. Mean pooling is an insufficient composition rule for reusable table-level representations, yet the backbone is not devoid of useful structure. The natural next question is whether this recoverable structure can be pushed further when table-level representation learning is built into training rather than added only after pretraining.

## 7 Proof-of-concept Source Retrieval: Parent-table Retrieval

To complement the proxy desiderata, we add a deliberately simple closed-corpus proof of concept. More than one downstream task could be used to probe the desiderata in practice, including retrieval, overlap-aware candidate generation, and approximate deduplication. We choose parent-table retrieval from partial views because it is simple, controlled, and representative of source-recovery settings in which a partial table must be matched back to its originating source or to a small set of near-duplicate candidates. We use the partition-to-parent setting, where each query is a sampled partition and the candidate corpus contains 100 source tables. Each embedder is evaluated on 1000 queries, and retrieval performance is reported at multiple cutoffs, $R@1$, $R@5$, and $R@10$, together with Mean Reciprocal Rank (MRR). This task is not intended to replace the benchmark or to claim open-world deployment realism. Rather, it tests whether the trends exposed by the benchmark and the frozen-backbone case study transfer to a concrete source-recovery setting. Because the query is a partial partition, success depends on D1-like stability across views of the same table, while the source-table target also makes it depend on the Direct-label component of D2.

Table 5 shows that the downstream trends largely mirror the benchmark. Parent-table retrieval rewards embeddings that keep partial views close to their source table while separating them from other source identities, which is precisely the combination of D1 partial-view consistency and Direct-label D2 discrimination. In this setting, schema- and content-sensitive fingerprints remain especially strong. On synthetic data, **HashingText** is dominant with near-perfect Recall@1 and MRR, while **Armadillo** is the strongest non-text baseline. On real data, **HashingSchema**, **HashingText**, and **LLMText** form the top group. This is consistent with the benchmark rather than in tension with it. When the target is source-table identity in a fixed corpus, schema and lexical cues are often operationally meaningful rather than spurious shortcuts.

The learned ConTextTab-R heads improve materially over pooled ConTextTab on the same task. On synthetic data, ConTextTab-R Attention reaches Recall@1 of 0.20 and MRR of 0.34, compared with 0.05 and 0.14 for pooled ConTextTab. On real data, ConTextTab-R GatedAttention reaches Recall@1 of 0.35 and MRR of 0.52, compared with 0.10 and 0.19 for pooled ConTextTab. These gains strengthen two claims of the paper at once. First, the benchmark is not merely diagnostic; it tracks behavior that matters for a downstream table-level operation. Second, useful table-level structure is recoverable from frozen encoder states by learning a better composition rule, even though the final performance remains bounded by the information already available in the backbone and still trails the strongest lightweight baselines.

## 8 Discussion

**The main empirical lesson is a representation-prediction tension.** Across both synthetic and real data, the benchmark consistently separates two regimes. Lightweight schema-, text-, and summary-based methods perform strongly on D1 and D2 under shared observation budgets and often dominate the practical quality-cost frontier, while pooled representations extracted from prediction-oriented TFMs are not reliably strong as table embeddings. This does not mean that TFMs are weak models. Rather, it shows that predictive competence does not automatically yield reusable table-level geometry after pooling. The central contribution of the benchmark is therefore diagnostic. It makes visible a failure mode that is easy to miss when tabular models are judged only by row-level prediction or by downstream tasks with task-specific supervision.

**What this suggests about compositionality.** The results support a compositional reading of table-level representation. Tables exhibit multiple interacting levels of structure, and the benchmark shows that preserving useful whole-table geometry depends not only on what information is present locally, but also on how that information is composed into a single representation. In this sense, the paper does not claim to solve compositional generalization in tabular learning. Instead, it shows that learned composition rules matter. Mean pooling is often too weak as a rule for turning local predictive states into reusable table-level structure, while better-aligned aggregation can recover a substantial part of the missing geometry from the same frozen encoder states.

**Why do simple methods work so well?** In many realistic corpus-scale operations, lexical and structural cues are not spurious shortcuts but part of the operational definition of similarity. Column names, type patterns, and recognizable value tokens often provide exactly the information needed for indexing, candidate generation, source recovery, and approximate deduplication. Retaining schema metadata in the benchmark is deliberate, as for many real-world tabular corpora, column titles and types are part of the available signal rather than an artificial advantage. This is why lightweight schema- and text-based methods remain so competitive in both the benchmark and the parent-table retrieval proof of concept. At the same time, the per-label D2 breakdown shows that these cues are not sufficient for every notion of similarity. Harder settings such as *Direct* identity ranking under hard negatives or *Semantic+Difficulty* labels expose the point at which shallow schema alignment stops being enough and value-sensitive structure becomes more important. The column-name anonymization ablation in Appendix A.3 makes this distinction explicit: schema-only methods degrade when headers are removed, while HASHINGTEXT and name-light statistical controls remain stable, showing that exact column-name matching is not the sole driver of the lightweight baselines' competitiveness.

**Why does the frozen-backbone case study matter?** The case study sharpens the diagnosis. The benchmark does not merely show that pooled TFMs lag behind lightweight baselines; it also shows that much of this lag is recoverable. Freezing CONTEXTTAB and training a lightweight, desiderata-aligned representation head yields substantial gains on D1 and D2, and those gains persist in the parent-table retrieval proof of concept. Since the encoder is frozen and the SCDM objective is aligned with D1/D2, this is a diagnostic recoverability study rather than a claim of universal open-world representation learning. As detailed in Appendix A.9, the heads are selected and evaluated under separate validation and final benchmark protocols, and Section 7 further tests them in parent-table retrieval rather than only under the SCDM training signal. First, the pooled weakness is not solely a backbone limitation; aggregation and supervision are genuine bottlenecks. Second, the gains remain bounded by the information ceiling of the frozen backbone, which explains why the learned heads improve substantially over pooled CONTEXTTAB without changing the underlying D4 efficiency regime.

**What does the downstream task add?** Parent-table retrieval is a simple downstream task, but it plays an important role as a proof of concept. More than one downstream task could be aligned with the desiderata, including overlap-aware candidate generation or approximate deduplication. We chose closed-corpus parent-table retrieval because it is controlled, directly interpretable, and representative of source-recovery scenarios under partial observations. Its behavior is consistent with the benchmark rather than independent of it. Methods that preserve partial-view stability and identity-sensitive discrimination under the benchmark also perform well in this retrieval setting, while the learned heads improve pooled CONTEXTTAB in the same

direction predicted by the case study. This supports the benchmark as a useful guide for model design without implying robustness to open-world retrieval or external distribution shift.

**Relation to tabular RAG and language agents.** Parent-table retrieval is not a full tabular RAG or language-agent benchmark; it isolates the table-selection layer that such systems must solve before an LLM can generate code, plan an analysis, or reason over retrieved evidence. In that sense, the task evaluates a representation-level prerequisite for tabular RAG rather than end-to-end agent performance. A complete pipeline would additionally mix in natural-language query interpretation, retrieval orchestration, re-ranking, prompting, program generation, and downstream reasoning, which are important but system-level factors beyond the representation question studied here.

**What does the benchmark establish, and what are its limitations?** The benchmark is designed to evaluate reusable table-level representations under a shared observation interface, not to exhaust every notion of useful table similarity. It does not test multi-table reasoning, open-world retrieval, or downstream tasks with richer semantic composition, and some robustness metrics saturate for methods that are permutation-invariant by construction. For that reason, we view the benchmark as a principled first layer rather than a complete final answer. Its role is to expose a concrete failure mode, provide a reproducible way to compare method families, and reveal whether table-level geometry is recoverable from frozen predictive backbones.

**Implications.** The main implication is that future tabular foundation models should be optimized not only for prediction, but also for reusable table-level geometry. There are at least two paths forward. One is to train native table-level interfaces directly. The other is to couple predictive objectives with explicit representation-level supervision so that the encoder is encouraged to preserve partial-view consistency and discriminability across multiple similarity notions during training itself. More broadly, the results suggest that table-level representation should be treated as a first-class design objective in tabular modeling rather than as an automatic byproduct of row-level predictive success.

## 9 Conclusion

We presented a unified evaluation framework for reusable table-level representations and used it to study how different table embedding methods behave under a shared observation interface. Across synthetic and real data, we found a consistent pattern. Lightweight schema- and text-based methods often outperform pooled embeddings from strong predictive tabular foundation models on the practical quality-cost frontier. This reveals a representation-prediction tension in current tabular foundation models. Strong row-level predictive performance does not by itself guarantee strong reusable table-level geometry after pooling. More broadly, our results suggest that useful table-level representations depend not only on what information is present in local predictive states, but also on how that information is composed into a global representation.

The paper is not only diagnostic. A case study on recoverability from a frozen CONTEXTTAB encoder shows that a substantial part of the gap can be recovered by learning a lightweight representation head aligned with the benchmark desiderata, indicating that aggregation and objective mismatch are plausible bottlenecks. At the same time, the gains remain bounded by the information already available in the frozen backbone. This suggests that predictive encoder states already contain useful local structure, but that turning this structure into a strong reusable table-level representation depends on learning an appropriate composition rule. A closed-corpus parent-table retrieval proof of concept then shows that these benchmark trends transfer to a concrete source-recovery setting, supporting the practical relevance of the benchmark for retrieval-oriented and corpus-scale table operations.

Taken together, these results suggest that reusable table-level geometry should be treated as a first-class design objective in future tabular foundation models rather than as an automatic byproduct of row-level predictive success. More broadly, we hope that the benchmark, the frozen-backbone case study, and the source-recovery proof of concept together provide a clearer foundation for evaluating and improving table representations for retrieval, overlap-aware candidate generation, approximate deduplication, curation, and other corpus-scale operations.

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

# A    Appendix

## A.1    Detailed descriptions of table embedding models

A table-level embedding model is a function $f$ that maps a table $T$ to a fixed-dimensional vector $f(T) \in \mathbb{R}^d$. To compare two tables under a fixed embedding method, we measure similarity between their corresponding embedding vectors. In this paper, we use cosine similarity as follows $\text{sim}(T_a, T_b) = \cos(f(T_a), f(T_b))$. Although embedding models conceptually operate on full tables, our evaluation applies $f(\cdot)$ to *table partitions* generated from the same source table, which are themselves valid subtables. This yields a unified interface for benchmarking diverse embedding families under controlled partial-views and perturbation regimes (Section A.2).

### A.1.1    Table embedding models

We evaluate table embedding models from multiple families with distinct inductive biases and input preprocessing. Methods that emit row- or cell-level representations are mapped to a single table vector via a fixed pooling operator. For serialization-based methods, we calculate a deterministic CSV-based string serialization $s(T)$ using a fixed number of rows. Unless stated otherwise, we apply mean pooling: for row outputs $L \in \mathbb{R}^{n \times d}$, with $L_i \in \mathbb{R}^d$ the $i$-th row vector, we obtain the final embedding as $f(T) = \frac{1}{n} \sum_{i=1}^{n} L_i$. For cell outputs $X \in \mathbb{R}^{n \times m \times d}$, with $X_{ij} \in \mathbb{R}^d$ a cell vector, we obtain the final embedding as $f(T) = \frac{1}{nm} \sum_{i=1}^{n} \sum_{j=1}^{m} X_{ij}$, where $n$ is the number of rows and $m$ is the number of columns. All embedding models are evaluated under the same observation interface, meaning that if a method requires a fixed input budget (e.g., due to token/row limits), the limits are applied to the same table view used elsewhere in evaluation. Hyperparameters are reported in Appendix A.5 (Table 8). Below, we list all table embedding models used in our experiments and group them into four distinct categories based on their underlying design principles.

**(A) Structural & statistical information.**    These methods are deterministic and emphasize schema and distributional signatures.

- **HashingSchema.** Schema tokens consist of column titles paired with inferred data types, serialized as $\tau(T) = \{(\text{col}_j, \text{dtype}_j)\}_{j=1}^{m}$, and mapped by a hashing vectorizer $h$ with $d$ bins (Pedregosa et al., 2011), yielding $f(T) = h(\tau(T))$.

- **SchemaContent.** Schema features are first serialized and hashed and subsequently concatenated with metadata features: $f(T) = [\phi_{\text{meta}}(T) \parallel h(\tau(T))]$, where $\phi_{\text{meta}}$ is a fixed-size vector that includes size, missingness, and type ratios.

- **TableStatistics.** A global feature vector $\phi(T)$ is formed from row/column counts, missingness ratios, average missing per row/column, and mean and standard deviation of per-column unique counts. Coarse type ratios (numeric/categorical/datetime) are appended, and the vector is padded or truncated to a fixed dimension.

- **StatisticalSummary.** For numeric columns $C_{\text{num}}$, per-column moments (min, max, mean, standard deviation, skew, median) are computed and aggregated by mean and standard deviation across columns. For categorical columns $C_{\text{cat}}$, cardinality, mode frequency, and entropy are computed and aggregated. For datetime columns $C_{\text{time}}$, min/max timestamps are aggregated. The concatenation yields $f(T)$.

- **MatrixFactorization.** Let $M_{\text{num}} \in \mathbb{R}^{n \times m'}$ be the numeric submatrix after mean imputation and column centering. Its singular values $\sigma_1 \geq \cdots \geq \sigma_k$ define $f(T) = [\sigma_1, \ldots, \sigma_k]$; if no numeric columns exist, a zero vector is returned.

**(B) Serialization-based text embeddings.**    These methods treat tables as sequences and encode them with hashing or text models.

- **HashingText.** The table is serialized to a CSV-style string $s(T)$ and embedded as $f(T) = h(s(T))$ using a hashing vectorizer $h$ (Pedregosa et al., 2011).

- **LLMText.** A sentence-transformer encoder $e(\cdot)$ (Reimers & Gurevych, 2019) is applied to the serialized table, $f(T) = e(s(T))$, mapping tabular content into a semantic text space.

**(C) Specialized representation methods.** These methods encode explicit relational structure beyond flat vectors or train a model to optimize a representation objective.

- **TableVectorizer (skrub data, 2023).** Each row is mapped by Skrub's mixed-type vectorizer $g(\cdot)$ to a feature vector; the table embedding is mean pooled, $f(T) = \frac{1}{n} \sum_i g(\text{row}_i)$, with padding or truncation to a fixed dimension.

- **HyTREL (Chen et al., 2023).** A hypergraph is constructed with cell nodes and hyperedges for the table, columns, and rows. The encoder returns hyperedge embeddings. We take the table-level hyperedge vector as $f(T)$. We employ the provided model checkpoint pretrained with contrastive and discriminative objectives.

- **Armadillo (Pugnaloni et al., 2025).** Tables are converted to a graph with row, column, and value nodes; each value node connects to its corresponding row and column. Initial node features are produced by a token/value encoder, and a GNN propagates them across the graph. The table embedding is the global mean pool of node representations, $f(T) = \frac{1}{|V|} \sum_{v \in V} z_v$. We use the provided model checkpoint pretrained to estimate the overlap ratio between tables.

**(D) Foundation model-derived embeddings.** We extract final-layer hidden states from pretrained state-of-the-art foundation models and pool them to obtain a table vector.

- **TabPFN (Hollmann et al., 2023).** After preprocessing to obtain $(X, y)$, we follow the extension proposed in Ye et al. (2025) to extract row-level embeddings $L \in \mathbb{R}^{n \times d}$ conditioned on a target column; then we aggregate to get the full table embedding $f(T) = \frac{1}{n} \sum_{i=1}^{n} L_i$. If a target is absent, a valid non-degenerate column is selected.

- **ConTextTab (Spinaci et al., 2025).** The encoder outputs a contextual grid $X \in \mathbb{R}^{n \times m \times d}$ over rows and columns for a dummy randomly generated target. The dummy target token is removed and the grid is pooled, $f(T) = \frac{1}{nm} \sum_{i=1}^{n} \sum_{j=1}^{m} X_{ij}$.

## A.2 Detailed Evaluation Protocol

To evaluate the desiderata mentioned in Section 3, we propose a unified framework relying on proxy tasks that do not require downstream fine-tuning. We evaluate embeddings on *table partitions* sampled as partial views of each source table, and attach multi-view supervision signals to probe different notions of similarity and grouping. All sampling and evaluation hyperparameters are fixed across methods and reported in the Appendix (Tables 9 and 10).

### A.2.1 Partition sampling and overlap

For each source table $T$ with $n$ rows and $m$ columns, we sample $K_p$ partitions under a fixed observation budget (maximum number of rows/columns) applied consistently across all methods.[1] We model a table as a typed relation $T = (R, C, X)$ with row index set $R = \{1, \ldots, n\}$, column index set $C = \{1, \ldots, m\}$, and cell values $X \in \mathcal{V}^{n \times m}$. A *partition* is a subtable induced by row and column subsets, $P = T[R', C']$. Let $\mathcal{P}(T) = \{P_1, \ldots, P_{K_p}\}$ denote the sampled partitions from $T$. We represent the extent of each partition by its coordinate set

$$\mathcal{S}(P) = \{(i, j) \mid i \in R', j \in C'\},$$

and define the overlap between two partitions $P_a$ and $P_b$ using the Intersection over Union (IoU):

$$\text{IoU}(P_a, P_b) = \frac{|\mathcal{S}(P_a) \cap \mathcal{S}(P_b)|}{|\mathcal{S}(P_a) \cup \mathcal{S}(P_b)|}.$$

---

[1]Exact budgets and sampling hyperparameters are reported in Appendix A.5.

This overlap signal induces a natural notion of similarity between partial views of the same table and directly supports D1. In the sampler, the overlap-ratio parameter does not fix a target IoU directly. Instead, it controls the forced-overlap portion of partition generation by reusing a fraction of the rows and columns from a previously sampled partition before filling the remaining entries at random. We intentionally retain schema and column titles in this protocol. In corpus-scale tabular workflows such as indexing, retrieval, source recovery, curation, and approximate deduplication, column names and coarse types are often available metadata and therefore constitute a realistic signal rather than hidden supervision.

### A.2.2 Labeling schemes

To assess different granularities of grouping, each partition $P$ is assigned labels under four schemes:

- **Direct:** Table identity, $y_{\mathrm{dir}}(P) = \mathrm{id}(T)$, i.e., all partitions coming from the same table share the same label.

- **Semantic:** High-level semantic labels, $y_{\mathrm{sem}}(P) = g(T)$, where $g$ encodes higher-level semantics from metadata or generative factors. For synthetic data, tuples of generative factors (e.g., `physics formula + density`) are mapped to distinct labels, while for real data tuples are formed from benchmark source and dataset identifier (e.g., `carte + nba draft`).

- **Semantic+Difficulty:** Semantic labels with an additional difficulty distinction for synthetic data (e.g., `physics formula + density + medium`), where the difficulty is a generative factor that controls the complexity of table values without altering other features.

- **Stat:** Coarse statistical signature, $y_{\mathrm{stat}}(P) = l(P)$, where $l$ returns the majority coarse column type if its fraction exceeds a predefined threshold. A partition label defaults to a *mixed label* if no coarse type fraction exceeds the threshold.

### A.2.3 Metrics

To evaluate the performance of the different embedding models, we present a set of evaluation metrics that align with the desiderata (Figure 1). Unless stated otherwise, retrieval, triplet accuracy, and the linear probe accuracy are computed under each labeling scheme (Direct/Semantic/Semantic+Difficulty/Stat). We report scores averaged over label types for label-dependent metrics (D2 metrics) in the main Tables 2 and 3, with extended per-label breakdown in Figures 2 and 3 in the Appendix A.6.

**D1: Partition consistency (IoU correlation).** We measure the Spearman correlation between the partition overlap $\mathrm{IoU}(P_a, P_b)$ and the embedding similarity $\cos(f(P_a), f(P_b))$ over within-table pairs $\{(P_a, P_b) \mid P_a, P_b \in \mathcal{P}(T), a \neq b\}$. A higher correlation indicates that the embedding space preserves the similarity structure induced by overlapping partial views. We additionally report Pearson correlation and mean-squared-error (MSE) in Figures 2 and 3.

**D2: Triplet ranking (TR).** Given an anchor partition $A$, a positive partition $B$ with the same label $(y(B) = y(A), B \neq A)$, and a negative partition $C$ with a different label $(y(C) \neq y(A))$, we evaluate whether the embedding places the positive closer to the anchor than the negative. We report the fraction of triplets satisfying $\cos(f(A), f(B)) > \cos(f(A), f(C))$ as a proxy for ranking capability. We report multiple versions of the triplet ranking depending on how the negative partition is sampled: **TR-R** (random negative), samples $C$ uniformly from the negative pool. **TR-H** (hardest negative), selects $C$ to be the most similar negative by cosine similarity. **TR-CH** (cluster-hard negative), selects $C$ as the most similar negative restricted to the anchor's MiniBatch $k$-means cluster (with a different label). We aggregate **TR-Avg** $= \frac{1}{3}(\mathrm{TR\text{-}R} + \mathrm{TR\text{-}H} + \mathrm{TR\text{-}CH})$, and report the different triplet metrics in Appendix A.6.

**D2: Clustering alignment (CL).** We cluster embeddings using MiniBatch $k$-means and evaluate alignment with labels using Purity, Normalized Mutual Information (NMI), and Adjusted Rand Index (ARI). We aggregate **CL-Avg** $= \frac{1}{3}(\mathrm{Purity} + \mathrm{NMI} + \mathrm{ARI})$, and report the different clustering metrics in Appendix A.6.

**D2: Retrieval (Recall@$K_r$).** For each anchor partition, we retrieve its $K_r$ nearest neighbors by cosine similarity, excluding the anchor itself. A retrieval query is successful if at least one of the top-$K_r$ neighbors shares the same label as the anchor. We report Recall@$K_r$ as the percentage of successful queries across all anchors.

**D2: Linear probe accuracy (LP).** We train a logistic regression classifier on top of the frozen partition embeddings to predict labels, assessing whether the embedding captures discriminative semantic or statistical information about the table.

**D3: Robustness.** We perturb each partition by (i) permuting rows and columns, (ii) masking a subset of cells, or (iii) adding small noise to cell values, then measure the cosine similarity between the embeddings of the original and perturbed partitions.

**D4: Efficiency (Throughput).** We report embedding throughput in tables per second (Tbl/s), measured as the wall-clock rate for computing embeddings over partitions measured on the same compute hardware. This enables comparison of the quality-cost trade-off across embedding methods.

## A.3 Column-name anonymization ablation

The main benchmark intentionally preserves column names because schema metadata is normally available in table discovery, indexing, source-recovery, and RAG-style retrieval workflows. In these settings, column names are not merely shortcuts; they are part of the observed table and often encode the operational meaning of a column. Still, the same signal can make an evaluation overly sensitive to lexical schema overlap, especially for schema-only and serialization-based methods. We therefore add a controlled column-name anonymization ablation to separate exact schema-name matching from content- and structure-based representation quality.

The ablation keeps the task fixed. We sample and label partitions exactly as in the main benchmark, then copy each sampled partition and rename its columns positionally to `col_1`, ..., `col_m` immediately before representation extraction. Cell values, inferred dtypes, labels, coordinate sets, and partition geometry are unchanged. The comparison therefore removes lexical column-name identity while preserving the same sampled views and evaluation targets. Tables 6 and 7 report changes from the original benchmark to this anonymized-column setting. D2 Avg denotes the mean of TR-Avg, CL-Avg, Recall@5, and LP; table entries are rounded for display, while rank deltas are recomputed over the common baseline methods. Positive rank deltas indicate worse relative rank under anonymization.

The ablation has a useful diagnostic property: it changes one input channel while keeping the sampled values and targets fixed. The implementation-level effect differs by method. HashingSchema serializes column names and dtypes, so anonymization directly removes its central lexical feature. SchemaContent combines name/type schema tokens with metadata, so only part of its input is changed. HashingText and LLMText serialize tables with headers and values, so anonymization removes the header channel but leaves cell content intact. TableStatistics, StatisticalSummary, and MatrixFactorization serve as name-light controls because they operate on size, type, value, or numeric summaries. For imported neural encoders that also represent row/column structure, we report the same deltas but interpret them more cautiously.

The outcome is mixed in an informative way. Schema cues clearly matter: HashingSchema drops sharply on synthetic data ($\Delta D1 = -0.29$, $\Delta D2$ Avg $= -0.12$) and also degrades on real data, especially in D2 ($\Delta D2$ Avg $= -0.18$). LLMText also loses D1 consistency under anonymization, suggesting that sentence-transformer similarity is sensitive to header lexical distribution for view-to-view matching, although its D2 Avg changes by only $-0.05$ in both corpora. At the same time, the strongest content-preserving lightweight method remains stable. HashingText changes by only $+0.01$ in D1 and $-0.01$ in D2 Avg on synthetic data, and by $-0.05$ in D1 and $-0.01$ in D2 Avg on real data. The name-light controls are also largely stable. Thus, schema information is a useful and realistic signal in table discovery settings, but the competitiveness of lightweight methods is not explained solely by exact column-name matching.

Table 6: Column-name anonymization deltas for synthetic data. Original values are taken from Table 2; anonymized values are computed after renaming partition columns to positional names. The input-channel column summarizes which implemented signal is most directly changed by anonymization. D2 Avg averages TR-Avg, CL-Avg, Recall@5, and LP. Positive ΔRank means worse rank after anonymization.

| Embedder | Input channel changed | D1 Spearman | | | D2 Avg | | | Summary | |
|---|---|---|---|---|---|---|---|---|---|
| | | Orig. | Anon. | Δ | Orig. | Anon. | Δ | ΔD3 Avg | ΔRank |
| HashingSchema | Column names + dtypes | 0.58 | 0.29 | -0.29 | 0.60 | 0.49 | -0.12 | 0.00 | +2.68 |
| SchemaContent | Names/types + metadata | 0.28 | 0.26 | -0.02 | 0.41 | 0.36 | -0.05 | 0.00 | +0.25 |
| TableStatistics | No names; size/type stats | 0.34 | 0.32 | -0.02 | 0.43 | 0.42 | 0.00 | 0.00 | -0.38 |
| StatisticalSummary | No names; value summaries | 0.39 | 0.37 | -0.02 | 0.57 | 0.57 | 0.00 | 0.00 | -0.33 |
| MatrixFactorization | No names; numeric values | 0.37 | 0.39 | +0.02 | 0.46 | 0.45 | 0.00 | 0.00 | -0.92 |
| HashingText | Full CSV headers + values | 0.54 | 0.55 | +0.01 | 0.73 | 0.72 | -0.01 | -0.01 | -0.42 |
| LLMText | Head-row CSV headers + values | 0.30 | -0.02 | -0.32 | 0.68 | 0.62 | -0.05 | 0.00 | +1.18 |
| TableVectorizer | Row values after vectorization | 0.32 | 0.33 | +0.01 | 0.38 | 0.38 | 0.00 | 0.00 | -0.75 |
| HyTREL | Headers + cell values | 0.18 | 0.16 | -0.02 | 0.54 | 0.53 | -0.01 | -0.01 | -0.03 |
| Armadillo | Column/value graph structure | 0.41 | 0.39 | -0.02 | 0.55 | 0.56 | 0.00 | 0.00 | -0.33 |
| TabPFN | Values + target choice | -0.13 | -0.13 | 0.00 | 0.41 | 0.41 | 0.00 | 0.00 | -0.25 |
| CONTEXTTAB | Column-name/content embeddings | 0.06 | -0.01 | -0.07 | 0.43 | 0.42 | 0.00 | 0.00 | -0.71 |

Table 7: Column-name anonymization deltas for real data. Original values are taken from Table 3; anonymized values are computed after renaming partition columns to positional names. The input-channel column summarizes which implemented signal is most directly changed by anonymization. D2 Avg averages TR-Avg, CL-Avg, Recall@5, and LP. Positive ΔRank means worse rank after anonymization.

| Embedder | Input channel changed | D1 Spearman | | | D2 Avg | | | Summary | |
|---|---|---|---|---|---|---|---|---|---|
| | | Orig. | Anon. | Δ | Orig. | Anon. | Δ | ΔD3 Avg | ΔRank |
| HashingSchema | Column names + dtypes | 0.32 | 0.25 | -0.07 | 0.69 | 0.50 | -0.18 | -0.01 | +1.04 |
| SchemaContent | Names/types + metadata | 0.26 | 0.25 | -0.01 | 0.49 | 0.37 | -0.12 | 0.00 | +0.38 |
| TableStatistics | No names; size/type stats | 0.22 | 0.21 | -0.01 | 0.42 | 0.41 | -0.01 | 0.00 | +0.17 |
| StatisticalSummary | No names; value summaries | 0.20 | 0.21 | +0.01 | 0.49 | 0.50 | +0.01 | 0.00 | -0.33 |
| MatrixFactorization | No names; numeric values | 0.17 | 0.19 | +0.02 | 0.33 | 0.33 | -0.01 | 0.00 | -0.29 |
| HashingText | Full CSV headers + values | 0.40 | 0.35 | -0.05 | 0.66 | 0.65 | -0.01 | 0.00 | -0.35 |
| LLMText | Head-row CSV headers + values | 0.14 | -0.02 | -0.16 | 0.70 | 0.65 | -0.05 | -0.01 | +0.85 |
| TableVectorizer | Row values after vectorization | 0.02 | 0.01 | -0.01 | 0.31 | 0.30 | -0.01 | 0.00 | -0.33 |
| HyTREL | Headers + cell values | 0.16 | 0.12 | -0.04 | 0.52 | 0.50 | -0.02 | -0.01 | -0.08 |
| Armadillo | Column/value graph structure | 0.21 | 0.22 | +0.01 | 0.53 | 0.53 | 0.00 | 0.00 | -0.67 |
| TabPFN | Values + target choice | 0.05 | 0.06 | +0.01 | 0.33 | 0.34 | +0.01 | 0.00 | -0.42 |
| CONTEXTTAB | Column-name/content embeddings | 0.18 | 0.18 | 0.00 | 0.48 | 0.46 | -0.02 | 0.00 | +0.04 |

## A.4 Data sources

We evaluate on both synthetic and real open-source corpora to balance controlled semantics with real-world variability. Across both settings, we sample 100 source tables and compute the metrics on $K_p = 10$ partitions per table under a shared observation budget (Table 10).

**Synthetic data.** Synthetic corpora are generated from predefined rules with controllable difficulty levels, enabling systematic variation in scale and nonlinearity while preserving table structure. Each generator family provides generative factors that induce higher-level groupings in the embedding space, which are used for semantic (and semantic+difficulty) labels defined in Section A.2. We define four dataset families:

- **Physics formulas.** Fixed schema with numeric relations (e.g., density, ideal gas, kinetic energy, Ohm's law) , emphasizing intra-column numerical reasoning.

- **Geometric shapes.** Mixed numeric/categorical tables with closed-form area/perimeter rules for different geometric shapes; additional random colors act as distractors.

- **Temporal reasoning.** Timestamped event sequences for process lifecycles (HR, order processing, invoicing). Each table is generated under a company-specific random profile with distinct evolution cycles.

- **SCM prior.** Data generated by structural causal models adapted from TabICL (Qu et al., 2025) implementation, with controlled complexity and regression/classification targets.

Difficulty levels control numeric ranges and the number of SCM layers, yielding progressively harder variations without changing the labeling interface.

**Real open-source data.** Real tables are sampled from CARTE (Kim et al., 2024), OpenML-CC18 (Bischl et al., 2021), and OpenML-CTR23 (Fischer et al., 2023). For each benchmark task, we form a table by concatenating the feature matrix with a single target column and evaluate on fixed-size partitions under the same budget as synthetic (Table 10). We retain task metadata (benchmark source, dataset identifier; e.g., `carte+nba_draft`) to derive semantic labels.

### A.5 Further experimental details

Table 8: Embedding model hyperparameters.

| Embedder | Dim | Key Hyperparameters |
|---|---|---|
| HashingSchema | 1024 | `n_features`=1024 (feature hashing on schema) |
| SchemaContent | 256 | `n_features`=256 |
| TableStatistics | 14 | `output_dim`=14 (fixed statistical features) |
| StatisticalSummary | 22 | Numeric: min, max, mean, std, skew, median; Categorical: cardinality, entropy |
| MatrixFactorization | 16 | `n_components`=16, `center`=True |
| HashingText | 1024 | `n_features`=1024 (feature hashing on text) |
| LLMText | 384 | `model`=all-MiniLM-L6-v2, `max_rows`=32 |
| TableVectorizer | 512 | `output_dim`=512, `cardinality_threshold`=10 |
| HyTREL | 768 | `model`=bert-base-uncased, `batch_size`=16 `max_token_len`=64, `max_col_len`=20, `max_row_len`=30 |
| Armadillo | 300 | `gnn_type`=GraphSAGE, `num_layers`=3 `hidden_channels`=300, `initial_embedding`=sha256 |
| TabPFN | 192 | `n_estimators`=1, `max_samples`=50k, `max_features`=2000 `max_classes`=10, `data source`=test |
| ConTextTab | 768 | `model_size`=base, `classification`=cross-entropy `regression`=l2, `num_regression_bins`=10 |

Table 9: Evaluation metrics and their parameters.

| Desideratum | Metric | Configuration |
|---|---|---|
| D1: Consistency | Pearson correlation Spearman correlation MSE | IoU vs. cosine similarity IoU vs. cosine similarity IoU vs. cosine similarity |
| D2: Triplet | TR-R (Random) TR-H (Hard) TR-CH (Cluster-Hard) TR-Avg | Random negative, margin=0.01 Hardest negative mining Hard negatives within clusters Mean of TR-R, TR-H, TR-CH |
| D2: Clustering | Purity NMI ARI CL-Avg | KMeans, n_cluster=10 Normalized mutual information Adjusted Rand index Mean of Purity, NMI, ARI |
| D2: Retrieval | Recall@5 | Top-5 nearest neighbors |
| D2: Linear Probe | Accuracy | Logistic regression (max_iter=1000) |
| D3: Robustness | Permutation Noise Masking | Row/column shuffle invariance $\sigma$=[0.0, 0.01, 0.05, 0.1] Gaussian noise [0,5,10,25]% random cell masking |
| D4: Efficiency | Tables/second | Embedding throughput (measured on Nvidia T4 GPU) |

Table 10: Data sampling and partitioning configuration.

| Parameter | Value | Parameter | Value | Parameter | Value |
|---|---|---|---|---|---|
| Tables | 100 | Partitions/table | 10 | Overlap ratio | 0.5 |
| Row fraction | [0.2, 0.5] | Column fraction | [0.2, 0.5] | Min partition | $10 \times 5$ |
| Seeds | 10 | Rows (synthetic) | 100 | Rows (real) | varies |

## A.6 Extended D2 per-label results

This appendix provides a label-wise decomposition of D2, complementing the label-averaged D2 scores reported in the main paper. For each label type, we report triplet ranking under three negative-sampling strategies (TR-R / TR-H / TR-CH), clustering alignment (Purity / NMI / ARI), retrieval (Recall@5), and linear-probe accuracy. TR-H and TR-CH are typically harder than TR-R because they focus on high-similarity negatives (hard negatives) and confusable negatives from the anchor's embedding cluster (cluster-hard negatives), respectively. The *Avg Rank* column summarizes performance across the individual D2 components for that label type.

### A.6.1 Synthetic data results

**Direct labeling exposes identity collisions under hard and cluster-hard negatives for most methods.** Under Direct labeling (Table 11), several methods achieve high TR-R yet collapse to near-zero on TR-H and TR-CH, confirming that random negatives are often trivial, while the hard and cluster-hard negatives reveal embedding collisions. HashingText is the clear outlier, remaining strong on both TR-H and TR-CH, consistent with preserving fine-grained identity features from serialized content. Notably, Armadillo is the only other method with non-trivial TR-H and TR-CH (while still trailing HashingText), indicating partial identity preservation beyond schema/statistical fingerprints. By contrast, schema-only and coarse statistical baselines (e.g., HashingSchema, TableStatistics, StatisticalSummary) retain reasonable TR-R but fail under hard negatives, consistent with many partitions sharing near-identical schema/type signatures. Finally, pooled TFM embeddings (TabPFN, ConTextTab) show moderate TR-R performance but collapse under hard negatives, suggesting that pooling produces weak identity geometry even when some coarse similarity is recoverable.

Table 11: D2 Metrics for Synthetic data - Direct Labels (mean ± 95% CI over 10 seeds)

| Embedder | Triplet (Direct) | | | | | Clustering (Direct) | | | | | Retrieval/Probe (Direct) | | | Overall |
|---|---|---|---|---|---|---|---|---|---|---|---|---|---|---|
| | TR-R | TR-H | TR-CH | TR-Avg | Rank | Purity | NMI | ARI | CL-Avg | Rank | R@5 | LP | Rank | Avg Rank |
| HashingSchema | 0.89±0.01 | 0.00±0.00 | 0.00±0.00 | 0.30±0.00 | 4.75 | 0.12±0.01 | 0.57±0.01 | 0.10±0.01 | 0.26±0.01 | 2.25 | 0.24±0.02 | 0.13±0.01 | 10.50 | 5.83 |
| SchemaContent | 0.02±0.00 | 0.00±0.00 | 0.00±0.00 | 0.01±0.00 | 11.75 | 0.07±0.00 | 0.33±0.01 | 0.03±0.00 | 0.14±0.00 | 10.25 | 0.36±0.01 | 0.10±0.01 | 10.00 | 10.67 |
| TableStatistics | 0.39±0.03 | 0.00±0.00 | 0.00±0.00 | 0.13±0.01 | 11.25 | 0.09±0.01 | 0.48±0.01 | 0.07±0.00 | 0.21±0.00 | 5.50 | 0.30±0.01 | 0.11±0.02 | 10.00 | 8.92 |
| StatisticalSummary | 0.80±0.01 | 0.00±0.00 | 0.00±0.00 | 0.27±0.00 | 4.50 | 0.10±0.01 | 0.52±0.01 | 0.08±0.00 | 0.23±0.00 | 4.25 | 0.73±0.01 | 0.15±0.02 | 4.50 | 4.42 |
| MatrixFactorization | 0.67±0.02 | 0.00±0.00 | 0.00±0.00 | 0.22±0.01 | 8.38 | 0.10±0.01 | 0.46±0.01 | 0.04±0.00 | 0.20±0.00 | 6.50 | 0.50±0.03 | 0.14±0.01 | 7.00 | 7.29 |
| HashingText | **0.98±0.00** | **0.67±0.02** | **0.68±0.02** | **0.78±0.01** | 1.00 | **0.13±0.01** | 0.56±0.01 | 0.08±0.01 | 0.26±0.01 | 2.50 | **0.99±0.00** | **0.99±0.00** | 1.00 | 1.50 |
| LLMText | 0.92±0.00 | 0.01±0.00 | 0.01±0.00 | 0.31±0.00 | 2.75 | 0.12±0.01 | **0.59±0.02** | **0.11±0.01** | **0.27±0.01** | 1.25 | 0.61±0.02 | 0.44±0.03 | 3.50 | 2.50 |
| TableVectorizer | 0.51±0.02 | 0.00±0.00 | 0.00±0.00 | 0.17±0.01 | 7.00 | 0.07±0.00 | 0.28±0.01 | 0.02±0.00 | 0.12±0.00 | 12.00 | 0.38±0.02 | 0.06±0.01 | 9.50 | 9.50 |
| HyTREL | 0.73±0.01 | 0.00±0.00 | 0.00±0.00 | 0.24±0.00 | 7.00 | 0.09±0.01 | 0.42±0.01 | 0.05±0.00 | 0.19±0.01 | 7.50 | 0.54±0.02 | 0.40±0.01 | 4.50 | 6.33 |
| Armadillo | 0.84±0.00 | 0.19±0.02 | 0.21±0.02 | 0.42±0.01 | 2.50 | 0.11±0.01 | 0.41±0.01 | 0.06±0.00 | 0.19±0.01 | 6.25 | 0.78±0.01 | 0.66±0.02 | 2.00 | 3.58 |
| TabPFN | 0.49±0.01 | 0.00±0.00 | 0.00±0.00 | 0.16±0.00 | 8.50 | 0.07±0.00 | 0.31±0.02 | 0.02±0.00 | 0.14±0.01 | 10.75 | 0.36±0.01 | 0.18±0.02 | 6.50 | 8.58 |
| ConTextTab | 0.54±0.01 | 0.00±0.00 | 0.00±0.00 | 0.18±0.00 | 8.62 | 0.08±0.01 | 0.34±0.01 | 0.04±0.00 | 0.15±0.01 | 9.00 | 0.27±0.01 | 0.14±0.02 | 9.00 | 8.88 |

**Semantic labeling highlights global grouping strength of serialization-based text embedding models.** Under Semantic labeling (Table 12), HashingSchema achieves the strongest triplet performance, implying that schema tokens correlate with the generative factors used to define the semantic labels. However, LLMText provides the best clustering alignment by a substantial margin, indicating that semantic text encoders better preserve the global grouping structure induced by the generators, beyond local ranking. Retrieval and linear probe scores are generally high in this regime, but remain discriminative, with HashingText reaching near-perfect retrieval and linear probe performance, reflecting that content-level fingerprints preserve both separability and linear decodability for these semantic groupings.

**Semantic+Difficulty labeling reveals schema-only limitations under value-complexity shifts.** Under Semantic+Difficulty labeling (Table 13), we refine semantic groups by adding a difficulty factor, so tables with the same schema can receive different labels depending on content complexity. Triplet scores drop broadly, indicating that many methods fail to distinguish partitions based on value-level complexity. The degradation is most pronounced for schema-centric hashing, with HashingSchema largely collapsing on TR-H and TR-CH, suggesting that difficulty is not recoverable from schema tokens alone and is instead expressed through value distributions and nonlinear relationships. In this harder setting, HashingText provides the strongest triplet ranking (including hard and cluster-hard negatives), whereas LLMText again achieves the best clustering alignment. This reinforces a recurring trade-off between content fingerprints

Table 12: D2 Metrics for Synthetic data - Semantic Labels (mean ± 95% CI over 10 seeds)

| Embedder | Triplet (Semantic) | | | | | Clustering (Semantic) | | | | | Retrieval/Probe (Semantic) | | | Overall |
|---|---|---|---|---|---|---|---|---|---|---|---|---|---|---|
| | TR-R | TR-H | TR-CH | TR-Avg | Rank | Purity | NMI | ARI | CL-Avg | Rank | R@5 | LP | Rank | Avg Rank |
| HashingSchema | **0.96±0.01** | **0.52±0.03** | **0.53±0.03** | **0.67±0.02** | 1.25 | **0.79±0.03** | 0.85±0.02 | 0.70±0.05 | 0.78±0.03 | 1.75 | 0.98±0.00 | 0.89±0.02 | 3.00 | 2.00 |
| SchemaContent | 0.02±0.00 | 0.00±0.00 | 0.00±0.00 | 0.01±0.00 | 11.75 | 0.38±0.01 | 0.43±0.01 | 0.19±0.01 | 0.34±0.01 | 9.25 | 0.98±0.00 | 0.88±0.01 | 4.50 | 8.50 |
| TableStatistics | 0.41±0.03 | 0.00±0.00 | 0.00±0.00 | 0.14±0.01 | 11.25 | 0.54±0.02 | 0.62±0.02 | 0.42±0.03 | 0.53±0.02 | 5.25 | 0.92±0.01 | 0.63±0.03 | 9.50 | 8.67 |
| StatisticalSummary | 0.83±0.01 | 0.02±0.00 | 0.02±0.00 | 0.29±0.01 | 5.25 | 0.60±0.02 | 0.66±0.01 | 0.46±0.03 | 0.58±0.02 | 4.00 | 0.97±0.00 | 0.69±0.02 | 6.50 | 5.25 |
| MatrixFactorization | 0.70±0.03 | 0.00±0.00 | 0.01±0.01 | 0.24±0.01 | 7.25 | 0.55±0.03 | 0.57±0.01 | 0.29±0.03 | 0.47±0.02 | 5.75 | 0.89±0.04 | 0.55±0.03 | 10.50 | 7.83 |
| HashingText | 0.80±0.01 | 0.12±0.01 | 0.14±0.01 | 0.35±0.01 | 3.25 | 0.61±0.03 | 0.71±0.02 | 0.53±0.04 | 0.62±0.03 | 3.00 | **1.00±0.00** | **0.99±0.00** | 1.00 | 2.42 |
| LLMText | **0.97±0.01** | 0.47±0.02 | 0.48±0.02 | 0.64±0.02 | 1.75 | **0.79±0.04** | **0.88±0.03** | **0.72±0.07** | **0.80±0.04** | 1.25 | 0.99±0.00 | 0.94±0.01 | 2.00 | 1.67 |
| TableVectorizer | 0.52±0.02 | 0.04±0.01 | 0.04±0.01 | 0.20±0.01 | 6.75 | 0.37±0.01 | 0.30±0.02 | 0.14±0.01 | 0.27±0.01 | 12.00 | 0.89±0.01 | 0.38±0.03 | 11.50 | 10.08 |
| HyTREL | 0.74±0.01 | 0.03±0.01 | 0.04±0.01 | 0.27±0.00 | 5.50 | 0.49±0.02 | 0.51±0.02 | 0.29±0.01 | 0.43±0.02 | 7.00 | 0.97±0.00 | 0.89±0.01 | 5.00 | 5.83 |
| Armadillo | 0.68±0.02 | 0.05±0.01 | 0.05±0.01 | 0.26±0.01 | 5.25 | 0.40±0.02 | 0.37±0.01 | 0.23±0.03 | 0.33±0.02 | 9.75 | 0.92±0.00 | 0.71±0.02 | 7.50 | 7.50 |
| TabPFN | 0.50±0.01 | 0.00±0.00 | 0.00±0.00 | 0.17±0.00 | 9.50 | 0.38±0.03 | 0.37±0.03 | 0.16±0.03 | 0.30±0.03 | 10.75 | 0.89±0.00 | 0.65±0.03 | 10.50 | 10.25 |
| ConTextTab | 0.55±0.01 | 0.00±0.00 | 0.00±0.00 | 0.18±0.00 | 9.25 | 0.42±0.03 | 0.39±0.02 | 0.23±0.03 | 0.35±0.02 | 8.25 | 0.92±0.01 | 0.79±0.02 | 6.50 | 8.00 |

which excel at fine-grained neighbor ranking, and serialization-based methods that stabilize global grouping. The Semantic+Difficulty labeling scheme supports one of the central narratives of this paper, namely that incorporating difficulty granularity into semantic labeling renders evaluation sensitive to whether embeddings capture numerical scale and complexity, as opposed to merely shallow schema-derived features.

Table 13: D2 Metrics for Synthetic data - Semantic+Difficulty Labels (mean ± 95% CI over 10 seeds)

| Embedder | Triplet (Semantic+Difficulty) | | | | | Clustering (Semantic+Difficulty) | | | | | Retrieval/Probe (Semantic+Difficulty) | | | Overall |
|---|---|---|---|---|---|---|---|---|---|---|---|---|---|---|
| | TR-R | TR-H | TR-CH | TR-Avg | Rank | Purity | NMI | ARI | CL-Avg | Rank | R@5 | LP | Rank | Avg Rank |
| HashingSchema | 0.91±0.01 | 0.00±0.00 | 0.00±0.00 | 0.31±0.00 | 4.25 | 0.40±0.03 | 0.70±0.02 | 0.37±0.04 | 0.49±0.03 | 2.00 | 0.64±0.04 | 0.44±0.02 | 8.00 | 4.75 |
| SchemaContent | 0.02±0.00 | 0.00±0.00 | 0.00±0.00 | 0.01±0.00 | 11.75 | 0.20±0.01 | 0.37±0.01 | 0.10±0.01 | 0.22±0.01 | 9.75 | 0.79±0.04 | 0.39±0.02 | 7.00 | 9.50 |
| TableStatistics | 0.40±0.03 | 0.00±0.00 | 0.00±0.00 | 0.13±0.01 | 11.25 | 0.27±0.01 | 0.53±0.01 | 0.21±0.01 | 0.34±0.01 | 5.75 | 0.64±0.02 | 0.31±0.02 | 11.50 | 9.50 |
| StatisticalSummary | 0.81±0.01 | 0.01±0.00 | 0.01±0.00 | 0.27±0.00 | 4.75 | 0.33±0.03 | 0.59±0.01 | 0.25±0.02 | 0.39±0.02 | 4.00 | 0.90±0.01 | 0.39±0.03 | 4.50 | 4.42 |
| MatrixFactorization | 0.68±0.02 | 0.00±0.00 | 0.00±0.00 | 0.23±0.01 | 7.75 | 0.32±0.02 | 0.53±0.01 | 0.16±0.02 | 0.34±0.02 | 5.25 | 0.75±0.03 | 0.40±0.03 | 6.50 | 6.50 |
| HashingText | 0.84±0.01 | **0.23±0.01** | **0.24±0.02** | **0.44±0.01** | 1.50 | 0.33±0.02 | 0.63±0.02 | 0.27±0.02 | 0.41±0.01 | 3.00 | **1.00±0.00** | **0.98±0.01** | 1.00 | 1.83 |
| LLMText | **0.93±0.00** | 0.01±0.01 | 0.01±0.01 | 0.32±0.01 | 2.25 | **0.41±0.03** | **0.72±0.02** | **0.38±0.04** | **0.50±0.03** | 1.00 | 0.90±0.00 | 0.72±0.02 | 2.50 | 1.92 |
| TableVectorizer | 0.51±0.02 | 0.01±0.00 | 0.01±0.00 | 0.18±0.01 | 6.50 | 0.20±0.01 | 0.27±0.02 | 0.07±0.01 | 0.18±0.01 | 11.50 | 0.70±0.02 | 0.21±0.02 | 10.00 | 9.33 |
| HyTREL | 0.72±0.01 | 0.00±0.00 | 0.00±0.00 | 0.24±0.00 | 7.00 | 0.26±0.01 | 0.45±0.01 | 0.16±0.01 | 0.29±0.01 | 7.00 | 0.84±0.01 | 0.68±0.02 | 4.00 | 6.00 |
| Armadillo | 0.70±0.01 | 0.07±0.01 | 0.08±0.01 | 0.29±0.00 | 3.50 | 0.23±0.01 | 0.36±0.01 | 0.14±0.01 | 0.24±0.01 | 8.25 | 0.85±0.01 | 0.62±0.02 | 4.00 | 5.25 |
| TabPFN | 0.48±0.01 | 0.00±0.00 | 0.00±0.00 | 0.16±0.00 | 8.50 | 0.19±0.01 | 0.32±0.02 | 0.08±0.02 | 0.20±0.02 | 11.25 | 0.66±0.01 | 0.34±0.04 | 9.50 | 9.75 |
| ConTextTab | 0.53±0.01 | 0.00±0.00 | 0.00±0.00 | 0.18±0.00 | 9.00 | 0.22±0.02 | 0.35±0.02 | 0.12±0.02 | 0.23±0.01 | 9.25 | 0.65±0.01 | 0.37±0.03 | 9.50 | 9.25 |

**Stat labeling shows that retrieval and linear probing for majority-type labels are near-perfect, leaving triplet and clustering to separate methods.** Under the Stat labeling scheme (Table 14), retrieval and linear probe are near-saturated for many methods, indicating that coarse statistical type signatures are easy to identify under the current partition budget. Therefore, triplet ranking and clustering become the primary discriminators in this regime. HashingSchema dominates overall, consistent with type- and schema-token features being sufficient for coarse statistical grouping. LLMText remains competitive, especially in triplet ranking, while content- and factorization-based embedding models vary more widely, reflecting that some methods preserve coarse type separability without forming clean global cluster geometry.

Table 14: D2 Metrics for Synthetic data - Stat Labels (mean ± 95% CI over 10 seeds)

| Embedder | Triplet (Stat) | | | | | Clustering (Stat) | | | | | Retrieval/Probe (Stat) | | | Overall |
|---|---|---|---|---|---|---|---|---|---|---|---|---|---|---|
| | TR-R | TR-H | TR-CH | TR-Avg | Rank | Purity | NMI | ARI | CL-Avg | Rank | R@5 | LP | Rank | Avg Rank |
| HashingSchema | **0.96±0.01** | **0.83±0.02** | **0.83±0.02** | **0.87±0.02** | 1.00 | **0.96±0.01** | **0.54±0.02** | **0.40±0.03** | **0.64±0.02** | 1.00 | **1.00±0.00** | **1.00±0.00** | 1.00 | 1.00 |
| SchemaContent | 0.02±0.00 | 0.00±0.00 | 0.00±0.00 | 0.01±0.00 | 12.00 | 0.77±0.03 | 0.04±0.02 | 0.03±0.01 | 0.28±0.01 | 11.00 | 1.00±0.00 | 0.99±0.01 | 2.00 | 8.33 |
| TableStatistics | 0.27±0.03 | 0.08±0.02 | 0.08±0.02 | 0.14±0.02 | 9.00 | 0.78±0.02 | 0.19±0.02 | -0.02±0.03 | 0.32±0.02 | 9.25 | 0.99±0.00 | 0.99±0.01 | 5.50 | 7.92 |
| StatisticalSummary | 0.87±0.02 | 0.47±0.02 | 0.48±0.04 | 0.61±0.02 | 3.00 | 0.94±0.01 | 0.43±0.03 | 0.37±0.05 | 0.58±0.03 | 2.75 | 0.99±0.00 | 0.99±0.01 | 5.75 | 3.83 |
| MatrixFactorization | 0.43±0.03 | 0.09±0.02 | 0.09±0.02 | 0.20±0.02 | 7.00 | 0.76±0.03 | 0.14±0.02 | -0.04±0.03 | 0.29±0.01 | 10.75 | 0.97±0.01 | 0.91±0.01 | 11.50 | 9.75 |
| HashingText | 0.56±0.01 | 0.14±0.02 | 0.25±0.05 | 0.32±0.02 | 4.50 | 0.91±0.05 | 0.40±0.11 | 0.29±0.12 | 0.53±0.09 | 4.00 | 1.00±0.00 | 0.98±0.01 | 5.50 | 4.67 |
| LLMText | 0.94±0.01 | 0.69±0.03 | 0.69±0.03 | 0.77±0.02 | 2.00 | **0.96±0.01** | 0.51±0.05 | 0.36±0.08 | 0.61±0.04 | 2.25 | 1.00±0.00 | 0.99±0.01 | 3.00 | 2.42 |
| TableVectorizer | 0.41±0.01 | 0.04±0.01 | 0.06±0.01 | 0.17±0.01 | 9.00 | 0.79±0.03 | 0.08±0.03 | -0.02±0.04 | 0.28±0.01 | 10.25 | 0.98±0.00 | 0.93±0.02 | 10.00 | 9.75 |
| HyTREL | 0.58±0.03 | 0.04±0.01 | 0.05±0.01 | 0.22±0.01 | 7.00 | 0.85±0.02 | 0.21±0.05 | 0.20±0.06 | 0.42±0.04 | 5.75 | 0.99±0.00 | 0.98±0.01 | 5.50 | 6.08 |
| Armadillo | 0.62±0.02 | 0.11±0.02 | 0.13±0.02 | 0.29±0.02 | 4.75 | 0.87±0.03 | 0.23±0.05 | 0.19±0.06 | 0.43±0.04 | 5.50 | 0.99±0.00 | 0.93±0.02 | 8.25 | 6.17 |
| TabPFN | 0.44±0.03 | 0.02±0.02 | 0.03±0.02 | 0.16±0.01 | 9.75 | 0.82±0.01 | 0.13±0.03 | 0.20±0.05 | 0.38±0.03 | 7.75 | 0.97±0.01 | 0.88±0.02 | 11.50 | 9.67 |
| ConTextTab | 0.46±0.01 | 0.02±0.00 | 0.04±0.02 | 0.17±0.01 | 9.00 | 0.82±0.04 | 0.17±0.09 | 0.09±0.09 | 0.36±0.07 | 7.75 | 0.98±0.00 | 0.96±0.01 | 8.50 | 8.42 |

**Averaging across labels hides specialization, but preserves the big picture.** Averaging over label types (Table 15) clarifies why no single method consistently performs best across D2 proxy tasks. LLMText achieves the best overall D2 ranking due to consistently strong clustering under the Semantic labeling scheme, while HashingText is the best on retrieval, linear probe and remains most reliable for Direct-style identity ranking under hard negatives. In contrast, HashingSchema ranks best on average triplet performance but is comparatively weaker on retrieval and linear probe, highlighting that schema fingerprints can match

semantic groupings without yielding robust instance-level neighborhoods. This decomposition explains the main-table D2 average scores, where different labeling schemes reflect different notions of table similarity, and Semantic+Difficulty specifically reveals where schema features stop being sufficient.

Table 15: Detailed D2 Metrics for Synthetic data - Averaged over Label Types (mean ± 95% CI over 10 seeds)

| Embedder | Triplet (Avg over Labels) | | | | | Clustering (Avg over Labels) | | | | | Retrieval/Probe (Avg over Labels) | | | Overall |
|---|---|---|---|---|---|---|---|---|---|---|---|---|---|---|
| | TR-R | TR-H | TR-CH | TR-Avg | Rank | Purity | NMI | ARI | CL-Avg | Rank | R@5 | LP | Rank | Avg Rank |
| HashingSchema | **0.93±0.01** | **0.34±0.01** | **0.34±0.02** | **0.54±0.01** | 1.25 | **0.57±0.02** | 0.67±0.02 | **0.39±0.03** | 0.54±0.02 | 1.75 | 0.72±0.02 | 0.61±0.01 | 7.50 | 3.50 |
| SchemaContent | 0.02±0.00 | 0.00±0.00 | 0.00±0.00 | 0.01±0.00 | 12.00 | 0.35±0.01 | 0.30±0.01 | 0.09±0.01 | 0.25±0.01 | 11.00 | 0.78±0.01 | 0.59±0.01 | 6.00 | 9.67 |
| TableStatistics | 0.37±0.03 | 0.02±0.00 | 0.02±0.00 | 0.14±0.01 | 9.75 | 0.42±0.02 | 0.45±0.01 | 0.17±0.02 | 0.35±0.01 | 5.75 | 0.71±0.01 | 0.51±0.02 | 10.50 | 8.67 |
| StatisticalSummary | 0.83±0.01 | 0.12±0.01 | 0.13±0.01 | 0.36±0.01 | 3.75 | 0.49±0.02 | 0.55±0.01 | 0.29±0.03 | 0.45±0.02 | 4.00 | 0.90±0.01 | 0.56±0.02 | 5.00 | 4.25 |
| MatrixFactorization | 0.62±0.03 | 0.02±0.01 | 0.02±0.01 | 0.22±0.01 | 7.00 | 0.44±0.02 | 0.42±0.01 | 0.12±0.02 | 0.32±0.01 | 6.75 | 0.78±0.03 | 0.50±0.02 | 9.00 | 7.58 |
| HashingText | 0.79±0.01 | 0.29±0.02 | 0.33±0.03 | 0.47±0.01 | 3.00 | 0.49±0.03 | 0.58±0.04 | 0.29±0.04 | 0.46±0.03 | 3.00 | **0.99±0.00** | **0.99±0.01** | 1.00 | 2.33 |
| LLMText | **0.94±0.01** | 0.29±0.02 | 0.30±0.01 | 0.51±0.01 | 2.00 | **0.57±0.02** | **0.67±0.03** | **0.39±0.05** | **0.55±0.03** | 1.25 | 0.87±0.01 | 0.77±0.01 | 3.00 | 2.08 |
| TableVectorizer | 0.49±0.02 | 0.02±0.01 | 0.03±0.01 | 0.18±0.01 | 7.25 | 0.36±0.01 | 0.23±0.02 | 0.06±0.01 | 0.21±0.01 | 11.75 | 0.74±0.01 | 0.40±0.02 | 10.00 | 9.67 |
| HyTREL | 0.69±0.01 | 0.02±0.01 | 0.02±0.01 | 0.25±0.01 | 7.25 | 0.42±0.01 | 0.40±0.02 | 0.17±0.02 | 0.33±0.02 | 6.00 | 0.83±0.01 | 0.74±0.01 | 4.00 | 5.75 |
| Armadillo | 0.71±0.01 | 0.11±0.01 | 0.12±0.01 | 0.31±0.01 | 5.00 | 0.40±0.02 | 0.34±0.02 | 0.15±0.02 | 0.30±0.02 | 7.75 | 0.88±0.01 | 0.73±0.02 | 3.50 | 5.42 |
| TabPFN | 0.48±0.02 | 0.01±0.00 | 0.01±0.01 | 0.16±0.01 | 10.00 | 0.36±0.02 | 0.28±0.02 | 0.11±0.03 | 0.25±0.02 | 10.25 | 0.72±0.01 | 0.51±0.03 | 9.00 | 9.75 |
| CONTEXTTAB | 0.52±0.01 | 0.00±0.00 | 0.01±0.00 | 0.18±0.00 | 9.75 | 0.38±0.02 | 0.31±0.04 | 0.12±0.03 | 0.27±0.03 | 8.75 | 0.70±0.01 | 0.57±0.02 | 9.50 | 9.33 |

### A.6.2 Real data results

Across open-source corpora, D2 scores remain strongly label-dependent and separate methods by *what notion of similarity they preserve*: instance identity (Direct), benchmark/task semantics (Semantic), or coarse type signatures (Stat). We therefore report label-wise D2 tables below and the averaged summary (Table 19) to make the specialization pattern explicit rather than relying on a single aggregated number.

**Direct labeling shows partial collapse for hard and cluster-hard negatives rather than the complete collapse.** Under Direct labeling scheme (Table 16), several methods attain non-trivial performance on TR-H and TR-CH, indicating that the identity task is challenging but not degenerate for open-source tables. Overall, LLMText attains the strongest TR-Avg and clustering scores, while HashingText remains among the best retrieval methods and gives the strongest linear-probe performance, consistent with content-aware signals being reliable for separating confusable tables. Schema-driven fingerprints (e.g., HashingSchema) remain competitive and substantially outperform purely statistical baselines under hard negatives, but still trail the content/serialization methods, suggesting that schema features alone cannot fully resolve similar tables that share column types and structural templates. Model-pooled embeddings (TabPFN, ConTextTab) remain weak on hard negatives and clustering, indicating that pooling yields coarse neighborhoods that are insufficient for strict identity ranking even when some separability exists.

Table 16: D2 Metrics for Real data - Direct Labels (mean ± 95% CI over 10 seeds)

| Embedder | Triplet (Direct) | | | | | Clustering (Direct) | | | | | Retrieval/Probe (Direct) | | | Overall |
|---|---|---|---|---|---|---|---|---|---|---|---|---|---|---|
| | TR-R | TR-H | TR-CH | TR-Avg | Rank | Purity | NMI | ARI | CL-Avg | Rank | R@5 | LP | Rank | Avg Rank |
| HashingSchema | 0.95±0.01 | 0.24±0.02 | 0.24±0.02 | 0.48±0.01 | 3.00 | 0.12±0.01 | 0.50±0.01 | 0.05±0.00 | 0.22±0.00 | 3.25 | 0.93±0.01 | 0.71±0.03 | 3.00 | 3.08 |
| SchemaContent | 0.37±0.03 | 0.00±0.00 | 0.00±0.00 | 0.12±0.01 | 11.25 | 0.08±0.01 | 0.32±0.03 | 0.02±0.01 | 0.14±0.01 | 10.25 | 0.64±0.02 | 0.71±0.03 | 5.50 | 9.00 |
| TableStatistics | 0.27±0.02 | 0.00±0.00 | 0.00±0.00 | 0.09±0.01 | 11.75 | 0.10±0.01 | 0.44±0.02 | 0.07±0.01 | 0.20±0.01 | 4.75 | 0.60±0.02 | 0.32±0.02 | 8.00 | 8.17 |
| StatisticalSummary | 0.81±0.01 | 0.06±0.01 | 0.07±0.01 | 0.32±0.01 | 5.00 | 0.11±0.01 | 0.45±0.02 | 0.07±0.01 | 0.21±0.01 | 3.75 | 0.73±0.03 | 0.30±0.02 | 7.50 | 5.42 |
| MatrixFactorization | 0.69±0.02 | 0.00±0.00 | 0.00±0.00 | 0.23±0.01 | 9.00 | 0.10±0.01 | 0.37±0.02 | 0.04±0.01 | 0.17±0.01 | 9.00 | 0.44±0.03 | 0.08±0.01 | 11.00 | 9.67 |
| HashingText | 0.96±0.01 | **0.37±0.03** | **0.39±0.03** | 0.57±0.02 | 1.75 | 0.12±0.01 | 0.49±0.01 | 0.06±0.01 | 0.22±0.01 | 3.50 | 0.95±0.01 | **0.77±0.04** | 1.50 | 2.25 |
| LLMText | **0.99±0.00** | **0.37±0.03** | 0.38±0.03 | **0.58±0.02** | 1.25 | **0.13±0.01** | **0.59±0.02** | **0.10±0.01** | **0.27±0.01** | 1.00 | **0.96±0.01** | 0.72±0.03 | 1.50 | 1.25 |
| TableVectorizer | 0.53±0.02 | 0.02±0.01 | 0.03±0.01 | 0.19±0.01 | 8.50 | 0.07±0.01 | 0.29±0.01 | 0.03±0.00 | 0.13±0.01 | 11.00 | 0.34±0.02 | 0.08±0.01 | 11.50 | 10.33 |
| HyTREL | 0.80±0.01 | 0.02±0.01 | 0.03±0.01 | 0.28±0.01 | 6.50 | 0.10±0.01 | 0.41±0.01 | 0.05±0.01 | 0.19±0.01 | 6.00 | 0.76±0.02 | 0.57±0.03 | 5.00 | 5.83 |
| Armadillo | 0.84±0.01 | 0.18±0.01 | 0.20±0.01 | 0.40±0.01 | 4.00 | 0.11±0.01 | 0.41±0.01 | 0.06±0.00 | 0.19±0.01 | 5.75 | 0.76±0.01 | 0.54±0.04 | 5.00 | 4.92 |
| TabPFN | 0.39±0.02 | 0.01±0.00 | 0.01±0.00 | 0.14±0.00 | 9.50 | 0.08±0.00 | 0.27±0.02 | 0.02±0.01 | 0.12±0.01 | 11.75 | 0.36±0.02 | 0.20±0.02 | 10.50 | 10.58 |
| CONTEXTTAB | 0.72±0.02 | 0.03±0.01 | 0.03±0.01 | 0.26±0.01 | 6.50 | 0.10±0.01 | 0.40±0.02 | 0.05±0.00 | 0.18±0.01 | 8.00 | 0.58±0.03 | 0.47±0.03 | 8.00 | 7.50 |

**Semantic labeling highlights the global grouping strength of serialization-based text embeddings for benchmark and task structure.** For Semantic labeling (Table 17), LLMText is the clear winner on both triplet ranking and clustering alignment, indicating that semantic encoders best preserve the global grouping induced by benchmark and task semantics. HashingText remains highly competitive and achieves near-top retrieval and linear probe, while schema-oriented methods (HashingSchema) retain strong triplet performance but weaker clustering, consistent with capturing coarse correlates of benchmark structure without producing globally coherent clusters.

Table 17: D2 Metrics for Real data - Semantic Labels (mean ± 95% CI over 10 seeds)

| Embedder | Triplet (Semantic) | | | | | Clustering (Semantic) | | | | | Retrieval/Probe (Semantic) | | | Overall |
|---|---|---|---|---|---|---|---|---|---|---|---|---|---|---|
| | TR-R | TR-H | TR-CH | TR-Avg | Rank | Purity | NMI | ARI | CL-Avg | Rank | R@5 | LP | Rank | Avg Rank |
| HashingSchema | 0.95±0.01 | 0.44±0.03 | 0.45±0.03 | 0.62±0.02 | 3.00 | 0.22±0.01 | 0.53±0.01 | 0.09±0.01 | 0.28±0.01 | 3.25 | 0.97±0.01 | 0.94±0.02 | 3.50 | 3.25 |
| SchemaContent | 0.37±0.03 | 0.00±0.01 | 0.00±0.01 | 0.13±0.01 | 10.50 | 0.13±0.01 | 0.32±0.03 | 0.04±0.01 | 0.16±0.02 | 10.25 | 0.77±0.02 | 0.95±0.01 | 5.00 | 8.58 |
| TableStatistics | 0.26±0.01 | 0.00±0.00 | 0.00±0.00 | 0.09±0.00 | 12.00 | 0.18±0.01 | 0.46±0.02 | 0.11±0.01 | 0.25±0.01 | 4.75 | 0.72±0.02 | 0.47±0.03 | 8.00 | 8.25 |
| StatisticalSummary | 0.82±0.01 | 0.11±0.02 | 0.11±0.02 | 0.35±0.01 | 5.00 | 0.20±0.02 | 0.47±0.02 | 0.11±0.01 | 0.26±0.01 | 3.75 | 0.82±0.03 | 0.41±0.02 | 7.50 | 5.42 |
| MatrixFactorization | 0.69±0.02 | 0.00±0.00 | 0.00±0.00 | 0.23±0.01 | 9.50 | 0.15±0.01 | 0.38±0.02 | 0.06±0.01 | 0.20±0.01 | 9.00 | 0.55±0.03 | 0.14±0.01 | 10.50 | 9.67 |
| HashingText | 0.96±0.00 | 0.61±0.02 | 0.65±0.02 | 0.74±0.02 | 2.00 | 0.23±0.01 | 0.52±0.02 | 0.10±0.02 | 0.28±0.01 | 3.25 | 0.98±0.00 | 0.99±0.01 | 1.50 | 2.25 |
| LLMText | **0.99±0.00** | **0.71±0.03** | **0.73±0.03** | **0.81±0.02** | 1.00 | **0.26±0.02** | **0.63±0.02** | **0.18±0.03** | **0.35±0.02** | 1.00 | **0.99±0.00** | 0.97±0.01 | 1.50 | 1.17 |
| TableVectorizer | 0.53±0.02 | 0.05±0.02 | 0.05±0.02 | 0.21±0.01 | 7.50 | 0.11±0.01 | 0.28±0.02 | 0.04±0.01 | 0.15±0.01 | 10.75 | 0.44±0.01 | 0.13±0.02 | 12.00 | 10.08 |
| HyTREL | 0.80±0.01 | 0.04±0.01 | 0.05±0.01 | 0.30±0.01 | 7.00 | 0.18±0.02 | 0.43±0.02 | 0.09±0.01 | 0.23±0.01 | 6.75 | 0.84±0.01 | 0.80±0.02 | 4.50 | 6.08 |
| Armadillo | 0.83±0.01 | 0.29±0.02 | 0.31±0.02 | 0.48±0.02 | 4.00 | 0.20±0.02 | 0.42±0.02 | 0.10±0.01 | 0.24±0.02 | 5.25 | 0.84±0.01 | 0.71±0.03 | 5.50 | 4.92 |
| TabPFN | 0.40±0.03 | 0.01±0.01 | 0.01±0.00 | 0.14±0.01 | 9.50 | 0.11±0.02 | 0.27±0.02 | 0.03±0.01 | 0.14±0.01 | 12.00 | 0.46±0.02 | 0.28±0.02 | 10.50 | 10.67 |
| ConTextTab | 0.73±0.02 | 0.04±0.01 | 0.05±0.01 | 0.27±0.02 | 7.00 | 0.17±0.01 | 0.41±0.02 | 0.08±0.01 | 0.22±0.01 | 8.00 | 0.71±0.03 | 0.70±0.02 | 8.00 | 7.67 |

**Stat labeling shows that retrieval and linear probing for majority-type labels are near-perfect, with schema features dominating the remaining signal.** For the Stat labeling scheme (Table 18), retrieval and linear probe accuracy are near-saturated for many methods, so the more discriminative signal comes from triplet and clustering. In this regime, HashingSchema dominates overall, consistent with majority-type grouping being largely determined by column-type and schema tokens, while LLMText remains the strongest non-schema alternative.

Table 18: D2 Metrics for Real data - Stat Labels (mean ± 95% CI over 10 seeds)

| Embedder | Triplet (Stat) | | | | | Clustering (Stat) | | | | | Retrieval/Probe (Stat) | | | Overall |
|---|---|---|---|---|---|---|---|---|---|---|---|---|---|---|
| | TR-R | TR-H | TR-CH | TR-Avg | Rank | Purity | NMI | ARI | CL-Avg | Rank | R@5 | LP | Rank | Avg Rank |
| HashingSchema | **0.73±0.02** | **0.50±0.01** | **0.53±0.01** | **0.58±0.01** | 1.00 | **0.83±0.01** | **0.38±0.12** | **0.46±0.12** | **0.56±0.09** | 1.00 | **1.00±0.00** | 0.95±0.01 | 2.00 | 1.33 |
| SchemaContent | 0.25±0.02 | 0.00±0.00 | 0.01±0.01 | 0.09±0.01 | 10.25 | 0.63±0.03 | 0.04±0.02 | -0.02±0.02 | 0.22±0.01 | 9.50 | **1.00±0.00** | 0.98±0.00 | 2.00 | 7.25 |
| TableStatistics | 0.14±0.01 | 0.00±0.00 | 0.00±0.00 | 0.05±0.00 | 12.00 | 0.63±0.03 | 0.06±0.04 | 0.01±0.01 | 0.23±0.02 | 7.75 | 0.98±0.00 | **0.99±0.00** | 4.50 | 8.08 |
| StatisticalSummary | 0.66±0.03 | 0.05±0.01 | 0.05±0.01 | 0.25±0.01 | 3.50 | 0.78±0.01 | 0.23±0.03 | 0.31±0.03 | 0.44±0.03 | 3.00 | 0.99±0.00 | 0.90±0.02 | 7.00 | 4.50 |
| MatrixFactorization | 0.47±0.01 | 0.00±0.00 | 0.00±0.00 | 0.16±0.00 | 9.00 | 0.61±0.03 | 0.01±0.01 | 0.01±0.01 | 0.21±0.01 | 10.75 | 0.96±0.01 | 0.63±0.02 | 11.50 | 10.42 |
| HashingText | 0.53±0.01 | 0.05±0.01 | 0.06±0.01 | 0.21±0.01 | 3.50 | 0.71±0.03 | 0.18±0.05 | 0.17±0.06 | 0.35±0.05 | 4.00 | **0.99±0.00** | 0.92±0.02 | 6.00 | 4.50 |
| LLMText | 0.67±0.02 | 0.10±0.02 | 0.10±0.03 | 0.29±0.02 | 2.00 | 0.81±0.04 | 0.31±0.07 | 0.40±0.09 | 0.51±0.07 | 2.00 | 0.99±0.00 | 0.95±0.01 | 3.50 | 2.50 |
| TableVectorizer | 0.42±0.02 | 0.01±0.00 | 0.01±0.00 | 0.15±0.01 | 8.00 | 0.63±0.02 | 0.04±0.02 | 0.02±0.02 | 0.23±0.01 | 7.00 | 0.95±0.00 | 0.70±0.04 | 11.50 | 8.83 |
| HyTREL | 0.45±0.02 | 0.01±0.00 | 0.01±0.00 | 0.16±0.01 | 8.00 | 0.62±0.03 | 0.02±0.02 | -0.01±0.01 | 0.21±0.01 | 10.75 | 0.98±0.00 | 0.93±0.02 | 6.50 | 8.42 |
| Armadillo | 0.51±0.01 | 0.02±0.00 | 0.02±0.00 | 0.18±0.00 | 5.25 | 0.65±0.02 | 0.02±0.03 | 0.01±0.03 | 0.23±0.02 | 8.00 | 0.97±0.00 | 0.83±0.02 | 9.00 | 7.42 |
| TabPFN | 0.36±0.02 | 0.00±0.00 | 0.00±0.00 | 0.12±0.01 | 9.75 | 0.70±0.03 | 0.11±0.03 | 0.15±0.04 | 0.32±0.03 | 5.00 | 0.96±0.00 | 0.79±0.02 | 10.00 | 8.25 |
| ConTextTab | 0.50±0.02 | 0.02±0.01 | 0.03±0.02 | 0.18±0.01 | 5.75 | 0.62±0.03 | 0.08±0.04 | -0.02±0.02 | 0.22±0.02 | 9.25 | **0.99±0.00** | 0.94±0.01 | 4.50 | 6.50 |

**Averaging over label types hides specialization, but still provides useful intuition.** When averaged over label types (Table 19), HashingSchema ranks best on average triplet ranking, LLMText ranks best via clustering alignment, and HashingText ranks best on retrieval and probe. This decomposition matches the interpretation in the main paper, as different labeling schemes lead to different similarity notions, so a single averaged D2 score can hide meaningful specialization.

Table 19: Detailed D2 Metrics for Real data - Averaged over Label Types (mean ± 95% CI over 10 seeds)

| Embedder | Triplet (Avg over Labels) | | | | | Clustering (Avg over Labels) | | | | | Retrieval/Probe (Avg over Labels) | | | Overall |
|---|---|---|---|---|---|---|---|---|---|---|---|---|---|---|
| | TR-R | TR-H | TR-CH | TR-Avg | Rank | Purity | NMI | ARI | CL-Avg | Rank | R@5 | LP | Rank | Avg Rank |
| HashingSchema | **0.88±0.01** | **0.39±0.02** | **0.41±0.02** | **0.56±0.02** | 1.25 | 0.39±0.02 | 0.47±0.04 | 0.20±0.04 | 0.35±0.04 | 2.00 | 0.97±0.01 | 0.87±0.02 | 3.50 | 2.25 |
| SchemaContent | 0.33±0.03 | 0.00±0.00 | 0.00±0.01 | 0.11±0.01 | 10.50 | 0.28±0.01 | 0.23±0.03 | 0.02±0.01 | 0.17±0.01 | 11.00 | 0.80±0.01 | 0.88±0.02 | 5.00 | 8.83 |
| TableStatistics | 0.22±0.01 | 0.00±0.00 | 0.00±0.00 | 0.07±0.00 | 12.00 | 0.30±0.02 | 0.32±0.02 | 0.06±0.01 | 0.23±0.01 | 5.50 | 0.77±0.02 | 0.59±0.02 | 8.00 | 8.50 |
| StatisticalSummary | 0.76±0.02 | 0.07±0.01 | 0.08±0.01 | 0.30±0.01 | 4.75 | 0.36±0.01 | 0.38±0.02 | 0.16±0.02 | 0.30±0.02 | 3.25 | 0.85±0.02 | 0.53±0.02 | 7.50 | 5.17 |
| MatrixFactorization | 0.61±0.02 | 0.00±0.00 | 0.00±0.00 | 0.21±0.01 | 9.50 | 0.29±0.02 | 0.25±0.02 | 0.04±0.01 | 0.19±0.01 | 9.25 | 0.65±0.02 | 0.28±0.02 | 11.00 | 9.92 |
| HashingText | 0.81±0.01 | 0.34±0.02 | 0.37±0.02 | 0.51±0.01 | 3.00 | 0.35±0.02 | 0.40±0.03 | 0.11±0.03 | 0.28±0.02 | 3.75 | **0.97±0.01** | **0.89±0.02** | 1.50 | 2.75 |
| LLMText | **0.88±0.01** | **0.39±0.03** | 0.40±0.03 | **0.56±0.02** | 1.75 | **0.40±0.02** | **0.51±0.03** | **0.23±0.04** | **0.38±0.03** | 1.00 | **0.98±0.00** | 0.88±0.02 | 1.50 | 1.42 |
| TableVectorizer | 0.49±0.02 | 0.03±0.01 | 0.03±0.01 | 0.18±0.01 | 8.00 | 0.27±0.01 | 0.20±0.02 | 0.03±0.01 | 0.17±0.01 | 11.75 | 0.58±0.02 | 0.30±0.03 | 11.50 | 10.42 |
| HyTREL | 0.68±0.01 | 0.03±0.01 | 0.03±0.01 | 0.24±0.01 | 7.00 | 0.30±0.02 | 0.29±0.02 | 0.04±0.01 | 0.21±0.01 | 7.25 | 0.86±0.01 | 0.77±0.02 | 4.50 | 6.25 |
| Armadillo | 0.73±0.01 | 0.16±0.01 | 0.18±0.01 | 0.36±0.01 | 4.25 | 0.32±0.02 | 0.28±0.02 | 0.06±0.02 | 0.22±0.02 | 6.50 | 0.86±0.01 | 0.69±0.03 | 6.00 | 5.58 |
| TabPFN | 0.39±0.02 | 0.01±0.00 | 0.01±0.00 | 0.13±0.01 | 9.50 | 0.29±0.01 | 0.22±0.02 | 0.06±0.02 | 0.19±0.02 | 8.75 | 0.60±0.01 | 0.42±0.02 | 10.50 | 9.58 |
| ConTextTab | 0.65±0.02 | 0.03±0.01 | 0.04±0.01 | 0.24±0.01 | 6.50 | 0.30±0.02 | 0.30±0.02 | 0.04±0.01 | 0.21±0.01 | 8.00 | 0.76±0.02 | 0.70±0.02 | 7.50 | 7.33 |

## A.7 Controlled D4-efficiency experiment

We report D4 separately from the D1-D3 quality tables because throughput is best measured in a controlled encode-only setting rather than mixed into quality rankings.

The controlled study evaluates six fixed scenarios obtained over three input modalities (numeric, text, mixed) with two table sizes. Small size (S) denotes a fixed $64 \times 16$ table and large size (L) denotes a fixed

$2048 \times 64$ table. The L condition is an intentional scaling stress test for encode-time behavior and is therefore substantially larger than the partition budget used in the main D1–D3 benchmark.

For each embedder and scenario, the evaluator generates 50 synthetic tables once, runs one untimed warm-up call to `encode_tables`, and then measures throughput over 5 timed encode-only repeats on the same 50 tables. We report throughput in tables per second (Tbl/s) together with 95% confidence intervals over repeats. As the main aggregate summary, we use the geometric mean (GeoMean) across the six scenario means. The Rank column in Table 20 is computed from GeoMean only.

Numeric tables contain normally distributed `float32` values, text tables contain short random lowercase ASCII strings of length 4–12, and mixed tables split the columns between numeric and text values. In particular, CONTEXTTAB and the learned CONTEXTTAB-R heads are evaluated with their own tokenizer and backbone, while LLMText is the separate sentence-transformer baseline and uses at most the first 32 rows. For serialization-based methods such as HashingText, numeric tables need not be faster than text tables because numeric `float32` values are serialized as decimal CSV strings, whereas the controlled text cells are short tokens.

Table 20 shows a clear order-of-magnitude spread in encode throughput. HashingSchema is the fastest method overall, followed by SchemaContent and TableStatistics. Prediction-oriented and graph-based methods are substantially slower under this controlled setup. For the learned heads on frozen CONTEXTTAB, we report all three representative variants used in the paper: Attention, GatedAttention, and Gated MLP. Their throughput remains very close to pooled CONTEXTTAB, showing that learned pooling improves representation quality while leaving the overall D4 efficiency largely unchanged.

Table 20: Controlled encode-only efficiency results for baseline embedders and learned CONTEXTTAB-R heads. Throughput is reported in tables per second (Tbl/s). S denotes $64 \times 16$ tables and L denotes $2048 \times 64$ tables. GeoMean is computed over the six scenario means, and Rank is computed from GeoMean only for the main baseline models. Baseline embedders are listed above the separator; learned heads are listed below it without ranks.

| Embedder | Numeric-S | Numeric-L | Text-S | Text-L | Mixed-S | Mixed-L | GeoMean Tbl/s | Rank |
|---|---|---|---|---|---|---|---|---|
| HashingSchema | 9793.76±112.47 | 2618.63±28.77 | 11675.20±77.04 | 3213.25±6.29 | 11694.96±911.74 | 3206.73±56.35 | 5748.43±104.02 | 1.00 |
| SchemaContent | 2293.53±17.80 | 928.61±2.98 | 1653.73±7.43 | 161.07±1.08 | 1765.45±31.58 | 270.29±4.26 | 804.30±3.58 | 2.00 |
| TableStatistics | 1471.32±15.68 | 201.71±4.73 | 1227.06±79.31 | 53.83±1.73 | 1201.78±10.75 | 87.43±3.62 | 356.70±8.17 | 3.00 |
| StatisticalSummary | 529.07±1.23 | 163.42±5.61 | 116.95±0.31 | 13.66±0.17 | 161.30±2.37 | 24.64±0.20 | 90.48±0.73 | 6.00 |
| MatrixFactorization | 368.22±11.62 | 72.43±0.26 | 360.33±9.88 | 73.04±0.91 | 355.93±1.53 | 74.27±0.50 | 162.71±0.96 | 5.00 |
| HashingText | 715.00±1.51 | 6.41±0.05 | 1265.68±1.79 | 12.26±0.04 | 900.99±3.22 | 8.34±0.13 | 90.10±0.34 | 7.00 |
| LLMText | 391.27±10.30 | 144.85±1.53 | 490.64±2.04 | 198.64±2.69 | 424.95±7.25 | 165.63±1.10 | 270.16±0.82 | 4.00 |
| TableVectorizer | 39.34±0.78 | 9.75±0.25 | 4.38±0.01 | 0.08±0.00 | 8.36±0.09 | 0.16±0.00 | 2.38±0.02 | 10.00 |
| HyTREL | 3.54±0.05 | 2.80±0.02 | 3.47±0.03 | 2.76±0.02 | 3.55±0.02 | 2.79±0.02 | 3.13±0.02 | 9.00 |
| Armadillo | 17.74±0.53 | 0.09±0.00 | 19.03±0.45 | 0.09±0.00 | 18.60±0.28 | 0.09±0.00 | 1.31±0.01 | 12.00 |
| TabPFN | 6.65±0.12 | 2.87±0.07 | 6.93±0.09 | 2.38±0.01 | 7.16±0.08 | 2.63±0.03 | 4.25±0.02 | 8.00 |
| CONTEXTTAB | 12.16±0.10 | 0.27±0.00 | 23.54±0.07 | 0.12±0.00 | 15.99±0.12 | 0.17±0.00 | 1.72±0.00 | 11.00 |
| *Learned heads on frozen CONTEXTTAB* | | | | | | | | |
| CONTEXTTAB-R Attention | 11.76±0.06 | 0.26±0.00 | 21.55±0.06 | 0.12±0.00 | 15.27±0.05 | 0.17±0.00 | 1.65±0.00 | - |
| CONTEXTTAB-R GatedAttention | 11.73±0.04 | 0.26±0.00 | 21.40±0.07 | 0.12±0.00 | 15.25±0.07 | 0.17±0.00 | 1.65±0.00 | - |
| CONTEXTTAB-R Gated MLP | 12.07±0.04 | 0.28±0.00 | 22.48±0.07 | 0.12±0.00 | 15.61±0.04 | 0.17±0.00 | 1.71±0.00 | - |

## A.8 Learned ConTextTab heads versus benchmark baselines

To quantify how much learned aggregation closes the gap between pooled tabular foundation-model embeddings and the strongest lightweight baselines, we insert the strongest representative CONTEXTTAB head from each family into the original leaderboard and recompute all ranks over the augmented method set. For synthetic data, we use the representative variants from the main paper: CONTEXTTAB + Attention ($\alpha = 0.75$), CONTEXTTAB + GatedAttention ($\alpha = 1.0$), and CONTEXTTAB + GatedMLP ($\alpha = 0.75$). For real data, we use CONTEXTTAB + Attention ($\alpha = 0.25$), CONTEXTTAB + GatedAttention ($\alpha = 0.5$), and CONTEXTTAB + GatedMLP ($\alpha = 1.0$). Ranks are recomputed exactly as in the main paper: D2 rank is the average rank over triplet accuracy, clustering, retrieval, and linear probing; D3 rank is the average rank over permutation, noise, and masking robustness; and the overall average rank is the average of D1, D2, and D3 ranks.

Table 21: Augmented synthetic-data leaderboard after adding the strongest representative CONTEXTTAB head from each family and recomputing all ranks over the enlarged method set (mean ± 95% CI over seeds). Lower ranks are better.

| Embedder | D1 Consistency | | D2 Label-Based - Avg | | | | | D3 Robustness | | | | Overall |
|---|---|---|---|---|---|---|---|---|---|---|---|---|
| | Spearman | Rank | TR-Avg | CL-Avg | R@5 | LP | Rank | Perm | Noise | Mask | Rank | Avg Rank |
| **ConTextTab + GatedAttention** ($\alpha = 1.0$) | 0.48 | 4.00 | 0.31 | 0.46 | 0.89 | 0.80 | 4.00 | 1.00 | 1.00 | 0.99 | 5.67 | 4.56 |
| **ConTextTab + Attention** ($\alpha = 0.75$) | 0.50 | 3.00 | 0.34 | 0.46 | 0.88 | 0.78 | 4.50 | 1.00 | 1.00 | 0.99 | 6.67 | 4.72 |
| HashingText | 0.54 | 2.00 | 0.47 | 0.46 | 0.99 | 0.99 | 2.50 | 1.00 | 0.66 | 0.97 | 10.00 | 4.83 |
| HashingSchema | 0.58 | 1.00 | 0.54 | 0.54 | 0.72 | 0.61 | 6.00 | 1.00 | 0.98 | 0.99 | 7.67 | 4.89 |
| **ConTextTab + GatedMLP** ($\alpha = 0.75$) | 0.48 | 5.00 | 0.30 | 0.43 | 0.88 | 0.80 | 5.25 | 1.00 | 1.00 | 0.99 | 5.67 | 5.31 |
| StatisticalSummary | 0.39 | 7.00 | 0.36 | 0.45 | 0.90 | 0.56 | 5.75 | 1.00 | 0.99 | 1.00 | 5.67 | 6.14 |
| MatrixFactorization | 0.37 | 8.00 | 0.22 | 0.32 | 0.78 | 0.50 | 11.00 | 1.00 | 1.00 | 1.00 | 2.67 | 7.22 |
| LLMText | 0.30 | 11.00 | 0.51 | 0.55 | 0.87 | 0.77 | 3.75 | 0.97 | 0.98 | 0.99 | 9.00 | 7.92 |
| Armadillo | 0.41 | 6.00 | 0.31 | 0.30 | 0.88 | 0.73 | 7.50 | 1.00 | 0.47 | 0.87 | 11.33 | 8.28 |
| TableStatistics | 0.34 | 9.00 | 0.14 | 0.35 | 0.71 | 0.51 | 12.25 | 1.00 | 1.00 | 1.00 | 4.00 | 8.42 |
| SchemaContent | 0.28 | 12.00 | 0.01 | 0.25 | 0.78 | 0.59 | 11.75 | 1.00 | 1.00 | 1.00 | 4.33 | 9.36 |
| HyTREL | 0.18 | 13.00 | 0.25 | 0.33 | 0.83 | 0.74 | 8.00 | 0.99 | 0.98 | 0.89 | 11.67 | 10.89 |
| TableVectorizer | 0.32 | 10.00 | 0.18 | 0.21 | 0.74 | 0.40 | 13.00 | 0.31 | 1.00 | 0.82 | 11.00 | 11.33 |
| **ConTextTab (pooled)** | 0.06 | 14.00 | 0.18 | 0.27 | 0.70 | 0.57 | 12.25 | 0.97 | 0.97 | 0.97 | 12.33 | 12.86 |
| TabPFN | -0.13 | 15.00 | 0.16 | 0.25 | 0.72 | 0.51 | 12.50 | 0.76 | 0.94 | 0.99 | 12.33 | 13.28 |

Table 22: Augmented real-data leaderboard after adding the strongest representative CONTEXTTAB head from each family and recomputing all ranks over the enlarged method set (mean ± 95% CI over seeds). Lower ranks are better.

| Embedder | D1 Consistency | | D2 Label-Based - Avg | | | | | D3 Robustness | | | | Overall |
|---|---|---|---|---|---|---|---|---|---|---|---|---|
| | Spearman | Rank | TR-Avg | CL-Avg | R@5 | LP | Rank | Perm | Noise | Mask | Rank | Avg Rank |
| HashingText | 0.40 | 1.00 | 0.51 | 0.28 | 0.97 | 0.89 | 2.75 | 1.00 | 0.75 | 0.98 | 8.50 | 4.08 |
| HashingSchema | 0.32 | 2.00 | 0.56 | 0.35 | 0.97 | 0.87 | 3.25 | 1.00 | 0.94 | 0.95 | 9.50 | 4.92 |
| **ConTextTab + Attention** ($\alpha = 0.25$) | 0.27 | 3.00 | 0.41 | 0.30 | 0.96 | 0.87 | 4.50 | 1.00 | 0.99 | 0.98 | 8.33 | 5.28 |
| **ConTextTab + GatedAttention** ($\alpha = 0.5$) | 0.26 | 4.00 | 0.38 | 0.27 | 0.95 | 0.87 | 5.50 | 1.00 | 0.99 | 0.99 | 6.67 | 5.39 |
| **ConTextTab + GatedMLP** ($\alpha = 1.0$) | 0.24 | 6.00 | 0.31 | 0.27 | 0.95 | 0.88 | 5.75 | 1.00 | 1.00 | 0.99 | 5.67 | 5.81 |
| SchemaContent | 0.26 | 5.00 | 0.11 | 0.17 | 0.80 | 0.88 | 10.25 | 1.00 | 1.00 | 1.00 | 4.00 | 6.42 |
| StatisticalSummary | 0.20 | 9.00 | 0.30 | 0.30 | 0.85 | 0.53 | 8.00 | 1.00 | 1.00 | 1.00 | 3.50 | 6.83 |
| TableStatistics | 0.22 | 7.00 | 0.07 | 0.23 | 0.77 | 0.59 | 11.25 | 1.00 | 1.00 | 1.00 | 2.83 | 7.03 |
| LLMText | 0.14 | 13.00 | 0.56 | 0.38 | 0.98 | 0.88 | 1.50 | 0.92 | 0.94 | 0.99 | 10.33 | 8.28 |
| MatrixFactorization | 0.17 | 11.00 | 0.21 | 0.19 | 0.65 | 0.28 | 12.75 | 1.00 | 1.00 | 1.00 | 3.17 | 8.97 |
| Armadillo | 0.21 | 8.00 | 0.36 | 0.22 | 0.86 | 0.69 | 8.25 | 1.00 | 0.59 | 0.77 | 11.17 | 9.14 |
| **ConTextTab (pooled)** | 0.18 | 10.00 | 0.24 | 0.21 | 0.76 | 0.70 | 10.50 | 0.97 | 0.97 | 0.97 | 11.00 | 10.50 |
| HyTREL | 0.16 | 12.00 | 0.24 | 0.21 | 0.86 | 0.77 | 8.50 | 0.99 | 0.89 | 0.96 | 12.00 | 10.83 |
| TabPFN | 0.05 | 14.00 | 0.13 | 0.19 | 0.60 | 0.42 | 13.25 | 0.78 | 0.97 | 0.97 | 11.00 | 12.75 |
| TableVectorizer | 0.02 | 15.00 | 0.18 | 0.17 | 0.58 | 0.30 | 14.00 | 0.33 | 0.98 | 0.84 | 12.33 | 13.78 |

On synthetic data, learned pooling moves CONTEXTTAB from the bottom of the leaderboard to the top of the quality-only augmented ranking, with CONTEXTTAB + GatedAttention and + Attention ranking first and second overall. On real data, the same trend holds, although the gains are smaller: the learned heads move clearly above pooled CONTEXTTAB, with the Attention head ranking behind only HashingText and HashingSchema.

## A.9 Training and validation details for learned representation heads

The learned representation heads reported in the paper follow the default real-data training configuration and are selected with a lightweight validation pass before full benchmark evaluation. Table 23 summarizes the differences between training, checkpoint selection, and final reporting.

The training objective is aligned with the desiderata, but it is not identical to the benchmark metrics. The head is trained with an IoU-based SOFTCE listwise objective over same-table partitions, while D1 is reported as the Spearman correlation between IoU and cosine similarity over within-table pairs. Likewise, the training loss uses Direct same-table identity (which can be obtained for free) only to define local support and, through local/global normalization, induces an implicit contrastive signal against other tables in the batch. It does not train directly for the full D2 suite used in evaluation. The metrics of triplet ranking, clustering alignment, retrieval, and linear probing are all measured only at evaluation time. The case study should therefore be read as testing how much stronger table-level structure can be recovered from frozen CONTEXTTAB states

Table 23: Training, validation, and final benchmark protocols for learned representation heads.

| Phase | Data source | Tables / partitions | Partition policy | Notes |
|---|---|---|---|---|
| Training | Real (training) | 16 tables per batch 8 partitions per table | Row/column fractions: 0.5/0.5 Minimum partition: $8 \times 4$ Overlap ratio: 0.7 | Train the head on top of a fixed encoder IoU-based SOFTCE listwise objective Same-table support defined by Direct identity |
| Validation | Real (validation) | 20 tables 5 partitions per table | Row/column fractions sampled from $[0.2, 0.5]$ Minimum partition: $10 \times 5$ Overlap ratio: 0.5 | Evaluator-based `val_score` Mean of D1 Spearman and averaged D2 sections Computed on the validation pass Used to select checkpoints |
| Benchmark | Real + synthetic (evaluation) | 100 tables 10 partitions per table | Row/column fractions sampled from $[0.2, 0.5]$ Minimum partition: $10 \times 5$ Overlap ratio: 0.5 | Full D1–D3 benchmark reporting Evaluation-time partitions Synthetic results reflect transfer from heads trained on real tables |

when the composition rule is trained under an IoU-based self-supervised objective that is aligned with, but not identical to, the benchmark.

The current setup uses separate dataset roots and separate samplers for optimization, checkpoint selection, and final reporting. In particular, training uses a fixed partition policy (0.5/0.5 row/column fractions, minimum partition $8 \times 4$, overlap ratio 0.7), whereas validation and final benchmark reporting use a different evaluation-time policy with row/column fractions sampled from $[0.2, 0.5]$, minimum partition $10 \times 5$, and overlap ratio 0.5. The final benchmark also includes synthetic evaluation although the heads are trained on real data only. We therefore read the learned-head study as a controlled recoverability experiment under a shifted evaluation protocol, where the head is trained on real data under one partitioning regime, then evaluated under different sampler settings and, in the final benchmark, also on synthetic data. This already shows that stronger table-level structure can be recovered from frozen CONTEXTTAB states beyond the exact training configuration. The broader question of how such composition rules generalize across unseen source tables, alternative partition regimes, and new data families is more naturally a training-time question for future foundation models than a frozen-head question alone. In that sense, the present case study serves as a concrete motivation for integrating table-level composition objectives directly into foundation-model training and then studying their generalization more systematically.

## A.10   Full ablation for learned representation heads

For completeness, we report the full $\alpha$-sweep for all representation-head families and both data regimes. The representative configurations shown in the main paper (Table 4) are selected, within each head family and data regime, by the lowest observed D1–D3 quality average rank. This section makes the selection transparent and to show that the main claim does not depend on a single hyperparameter setting.

The full sweep reveals three consistent patterns. First, all learned heads improve over the pooled CONTEXTTAB baseline on D1 and D2 across a broad range of $\alpha$ values. Second, the best trade-off rarely occurs at the same $\alpha$ for all metrics: D1 often prefers a stronger local component, while D2 benefits from a mixture of local and global supervision. Third, the architectural differences are systematic rather than incidental: Attention tends to peak on D1 and often provides the best overall quality ranking, while GatedAttention and GatedMLP remain competitive on retrieval, linear probing, and robustness.

## A.11   Code and implementation details

The implementation is organized as an extensible benchmark harness. A shared sampler materializes partial-table views with coordinate sets and multi-granularity labels, and each embedding method implements the same `encode_tables` interface. The main `Evaluator` computes D1–D3 metrics from shared sampled partitions, the `EfficiencyEvaluator` isolates D4 through encode-only throughput measurements, and the `DownstreamEvaluator` implements the closed-corpus parent-table retrieval proof of concept. For the learned CONTEXTTAB-R heads, the code uses separate training and evaluation dataset roots, batched fixed-shape partitions, vectorized pairwise similarity/overlap computations, Hydra/OmegaConf configurations, checkpoint evaluation, and ablation rendering.

| Head | $\alpha$ | D1 Consistency | D2 Label-Based - Avg | | | | D3 Robustness | | |
|---|---|---|---|---|---|---|---|---|---|
| | | Spearman | TR-Avg | CL-Avg | R@5 | LP | Perm | Noise | Mask |
| CONTEXTTAB (pooled) | – | 0.06±0.02 | 0.18±0.00 | 0.27±0.03 | 0.70±0.01 | 0.57±0.02 | 0.97±0.00 | 0.97±0.00 | 0.97±0.00 |
| Attention | 0.00 | 0.44±0.02 | 0.35±0.01 | 0.46±0.03 | 0.87±0.01 | 0.78±0.02 | **1.00±0.00** | **0.99±0.00** | 0.98±0.00 |
| | 0.25 | 0.46±0.02 | 0.35±0.01 | 0.44±0.01 | 0.87±0.01 | 0.78±0.02 | **1.00±0.00** | **0.99±0.00** | **0.99±0.00** |
| | 0.50 | 0.45±0.02 | 0.34±0.01 | **0.47±0.02** | 0.88±0.01 | 0.78±0.02 | **1.00±0.00** | **1.00±0.00** | **0.99±0.00** |
| | 0.75 | **0.50±0.02** | 0.34±0.01 | 0.46±0.02 | 0.88±0.01 | 0.78±0.02 | **1.00±0.00** | **1.00±0.00** | **0.99±0.00** |
| | 1.00 | 0.47±0.02 | 0.28±0.01 | 0.46±0.02 | 0.87±0.01 | 0.73±0.02 | **1.00±0.00** | **1.00±0.00** | **0.99±0.00** |
| GatedAttention | 0.00 | 0.47±0.02 | 0.34±0.01 | 0.44±0.03 | 0.88±0.01 | 0.79±0.02 | **1.00±0.00** | **0.99±0.00** | **0.99±0.00** |
| | 0.25 | 0.45±0.02 | 0.35±0.01 | 0.43±0.04 | 0.88±0.01 | 0.79±0.02 | **1.00±0.00** | **0.99±0.00** | 0.98±0.00 |
| | 0.50 | 0.48±0.02 | **0.37±0.01** | 0.46±0.02 | 0.89±0.01 | 0.80±0.02 | **1.00±0.00** | **0.99±0.00** | **0.99±0.00** |
| | 0.75 | 0.43±0.02 | 0.34±0.01 | 0.45±0.03 | 0.88±0.01 | **0.80±0.02** | **1.00±0.00** | **0.99±0.00** | **0.99±0.00** |
| | 1.00 | 0.48±0.02 | 0.31±0.01 | 0.46±0.03 | **0.89±0.01** | **0.80±0.02** | **1.00±0.00** | **1.00±0.00** | **0.99±0.00** |
| GatedMLP | 0.00 | 0.46±0.02 | 0.33±0.01 | **0.47±0.02** | **0.89±0.01** | **0.80±0.02** | **1.00±0.00** | **0.99±0.00** | **0.99±0.00** |
| | 0.25 | 0.46±0.02 | 0.33±0.01 | 0.44±0.04 | **0.89±0.01** | **0.80±0.02** | **1.00±0.00** | **0.99±0.00** | **0.99±0.00** |
| | 0.50 | 0.47±0.02 | 0.34±0.01 | 0.44±0.05 | 0.88±0.01 | 0.79±0.02 | **1.00±0.00** | **1.00±0.00** | **0.99±0.00** |
| | 0.75 | 0.48±0.02 | 0.30±0.01 | 0.43±0.04 | 0.88±0.01 | **0.80±0.02** | **1.00±0.00** | **1.00±0.00** | **0.99±0.00** |
| | 1.00 | 0.46±0.02 | 0.31±0.01 | 0.45±0.04 | **0.89±0.01** | **0.80±0.02** | **1.00±0.00** | **1.00±0.00** | **0.99±0.00** |

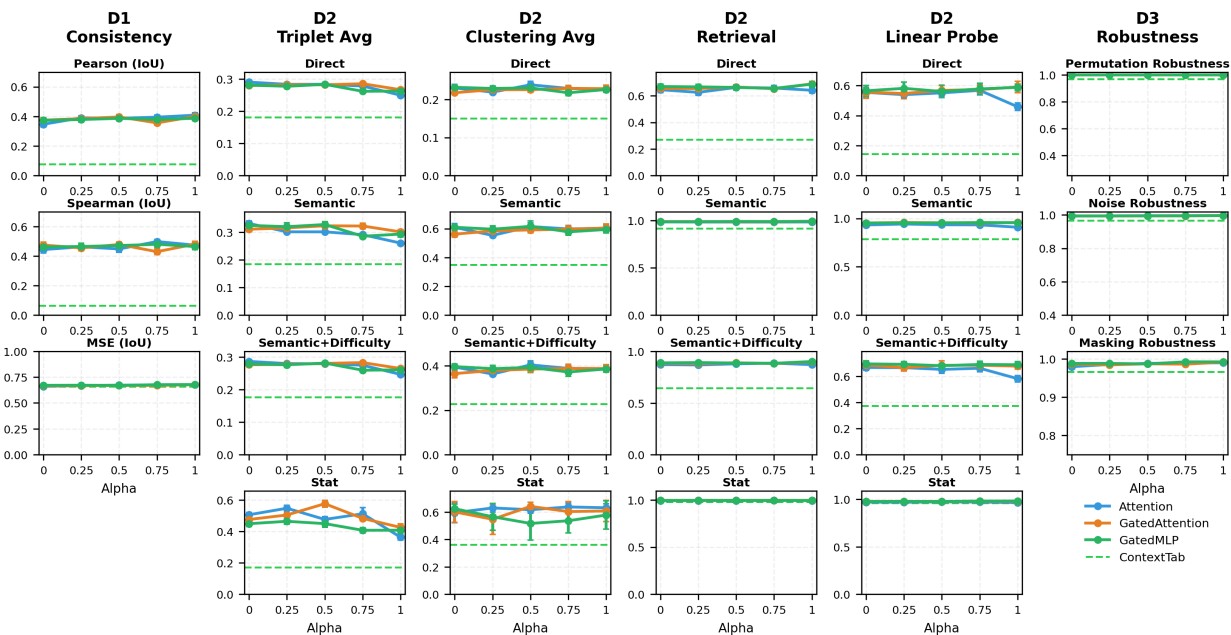

Table 24: Full Synthetic-data ablation for learned representation heads on top of frozen CONTEXTTAB. Top: complete $\alpha$-sweep table. Bottom: metric grid showing performance as a function of $\alpha$. The dashed green line corresponds to the pooled CONTEXTTAB baseline.

| Head | $\alpha$ | D1 Consistency | D2 Label-Based - Avg | | | | D3 Robustness | | |
|------|----------|----------------|---------|---------|---------|---------|---------|---------|---------|
| | | Spearman | TR-Avg | CL-Avg | R@5 | LP | Perm | Noise | Mask |
| CONTEXTTAB (pooled) | – | 0.18±0.01 | 0.24±0.01 | 0.21±0.01 | 0.76±0.02 | 0.70±0.02 | 0.97±0.00 | 0.97±0.00 | 0.97±0.00 |
| Attention | 0.00 | 0.26±0.03 | 0.40±0.02 | 0.29±0.03 | **0.96±0.01** | 0.86±0.02 | **1.00±0.00** | 0.99±0.01 | 0.98±0.00 |
| | 0.25 | **0.27±0.03** | 0.41±0.02 | 0.30±0.03 | **0.96±0.01** | 0.87±0.02 | **1.00±0.00** | 0.99±0.00 | 0.98±0.00 |
| | 0.50 | 0.25±0.03 | **0.42±0.02** | 0.31±0.04 | **0.96±0.00** | 0.86±0.02 | **1.00±0.00** | **0.99±0.00** | 0.98±0.00 |
| | 0.75 | 0.23±0.03 | 0.37±0.02 | 0.27±0.02 | 0.95±0.01 | 0.86±0.02 | **1.00±0.00** | **0.99±0.00** | **0.99±0.00** |
| | 1.00 | 0.17±0.03 | 0.24±0.01 | 0.28±0.04 | 0.91±0.01 | 0.83±0.02 | **1.00±0.00** | **1.00±0.00** | **1.00±0.00** |
| GatedAttention | 0.00 | 0.26±0.03 | 0.41±0.02 | 0.30±0.03 | **0.95±0.01** | 0.87±0.02 | **1.00±0.00** | **0.99±0.00** | 0.98±0.00 |
| | 0.25 | 0.23±0.03 | **0.42±0.02** | **0.34±0.05** | **0.95±0.01** | 0.87±0.02 | **1.00±0.00** | 0.99±0.00 | 0.98±0.00 |
| | 0.50 | 0.26±0.03 | 0.38±0.02 | 0.27±0.02 | 0.95±0.01 | **0.87±0.02** | **1.00±0.00** | **0.99±0.00** | 0.99±0.00 |
| | 0.75 | 0.27±0.04 | 0.39±0.02 | 0.30±0.04 | **0.95±0.01** | 0.87±0.02 | **1.00±0.00** | **0.99±0.00** | 0.98±0.00 |
| | 1.00 | 0.21±0.03 | 0.32±0.02 | 0.29±0.04 | 0.95±0.01 | **0.87±0.02** | **1.00±0.00** | **1.00±0.00** | **0.99±0.00** |
| GatedMLP | 0.00 | 0.24±0.04 | 0.41±0.02 | 0.28±0.02 | **0.96±0.01** | 0.87±0.02 | **1.00±0.00** | **0.99±0.00** | 0.99±0.00 |
| | 0.25 | 0.21±0.04 | 0.41±0.02 | 0.30±0.04 | 0.95±0.01 | 0.87±0.02 | **1.00±0.00** | **0.99±0.00** | 0.99±0.00 |
| | 0.50 | 0.23±0.03 | 0.40±0.02 | 0.27±0.02 | **0.96±0.01** | **0.88±0.02** | **1.00±0.00** | **0.99±0.00** | 0.99±0.00 |
| | 0.75 | 0.23±0.03 | 0.37±0.02 | 0.30±0.03 | 0.94±0.01 | 0.87±0.02 | **1.00±0.00** | **0.99±0.00** | **0.99±0.00** |
| | 1.00 | 0.24±0.03 | 0.31±0.02 | 0.27±0.03 | 0.95±0.01 | **0.88±0.02** | **1.00±0.00** | **1.00±0.00** | **0.99±0.00** |

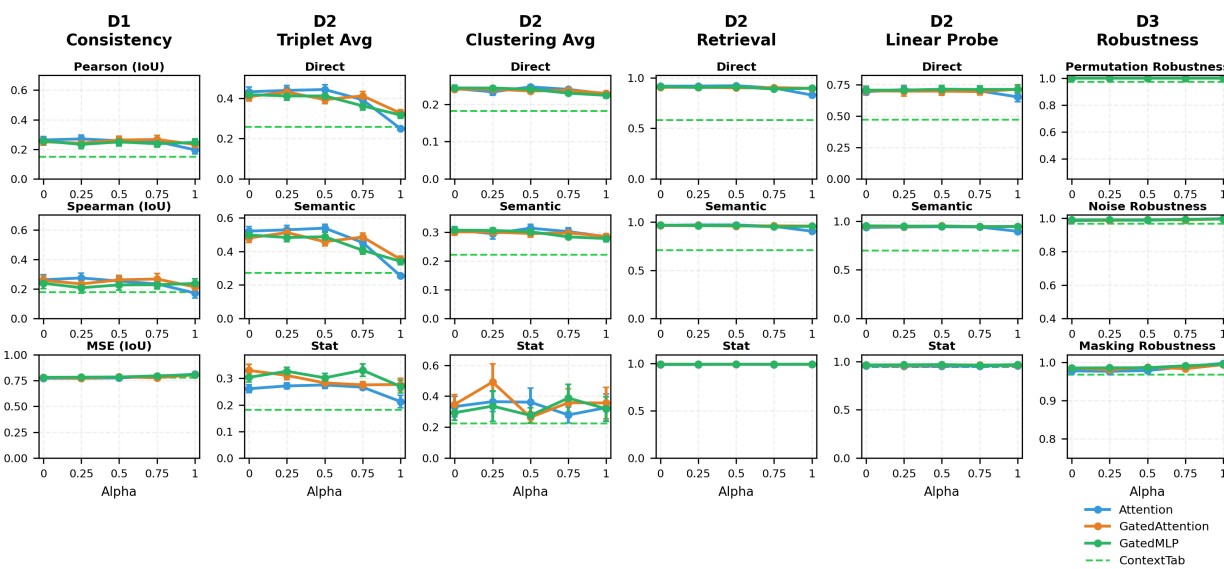

Table 25: Full Real-data ablation for learned representation heads on top of frozen CONTEXTTAB. Top: complete $\alpha$-sweep table. Bottom: metric grid showing performance as a function of $\alpha$. The dashed green line corresponds to the pooled CONTEXTTAB baseline.

We provide an anonymized ZIP package with the benchmark code, configurations, reproduction scripts, and table/figure rendering utilities as supplementary material; a clean public repository will be released upon acceptance.

### A.12 Large language model (LLM) usage disclosure

We used language models (mainly ChatGPT from OpenAI) to improve grammar and clarity of the manuscript and to suggest code refactorings during implementation. All benchmark design decisions, experiments, results, and conclusions are the authors' own; all LLM outputs were reviewed and validated, and the authors take full responsibility for the content.

### A.13 T-SNE plots

We provide t-SNE projections of partition embeddings as a qualitative sanity check for the label-wise D2 decomposition (Appendix A.6). Each row corresponds to an embedding model, and columns show the same embeddings colored by different labeling schemes. We emphasize that t-SNE is not metric-faithful; global distances are distorted and apparent cluster sizes are not comparable across methods. Nevertheless, label-homogeneous neighborhoods and visually coherent group structures often mirror the separability trends measured by triplet and clustering scores in the per-label D2 tables.

**Synthetic.** For synthetic corpora (Figure 5), the t-SNE projections qualitatively mirror the label-type dependence observed in the D2 decomposition (Appendix A.6). Under Direct labeling, methods that sustain hard-negative identity ranking (notably HashingText) exhibit more locally label-consistent neighborhoods, whereas pooled TFM representations (e.g., TabPFN, ConTextTab) form mixed-color manifolds with limited fine-grained separation. Under Semantic labeling, schema- and text-driven methods yield clearer group structure, aligning with their higher clustering scores. Crucially, moving from Semantic to Semantic+Difficulty visibly reduces separability for schema-centric fingerprints (e.g., HashingSchema), consistent with the Semantic+Difficulty triplet degradation in Table 13; this supports the interpretation that the added difficulty factor is weakly expressed in schema tokens and is instead reflected in value distributions and interaction patterns. We therefore treat t-SNE strictly as a qualitative diagnostic to support the quantitative D2 conclusions reported in Tables 11–13.

**Real open-source.** For open-source tables (Figure 6), the qualitative structure mirrors the D2 label dependence in Tables 16–18. Under Stat labeling, many methods exhibit visually separated regions, consistent with the relative ease of coarse type grouping. Under Direct labels, most embedding models show substantial color mixing, consistent with stricter identity ranking remaining challenging. Triplet ranking under hard and cluster-hard negatives exposes near-collisions even when retrieval and linear probe can be high. Semantic labels typically sit between these extremes, with semantic- and content-driven methods showing clearer group structure than purely schema- and statistical baselines.

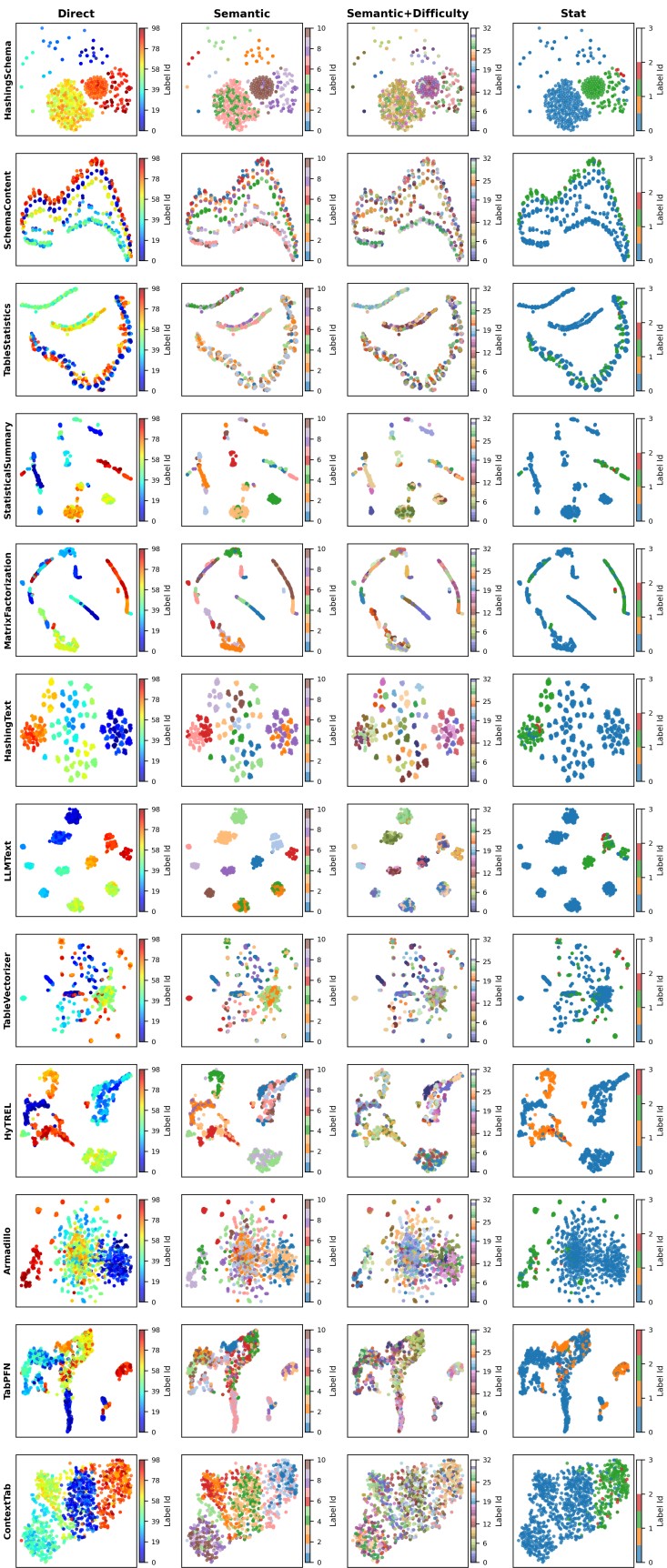

Figure 5: t-SNE projections of partition embeddings for synthetic data, colored by different labeling schemes (Direct, Semantic, Semantic+Difficulty, Stat).

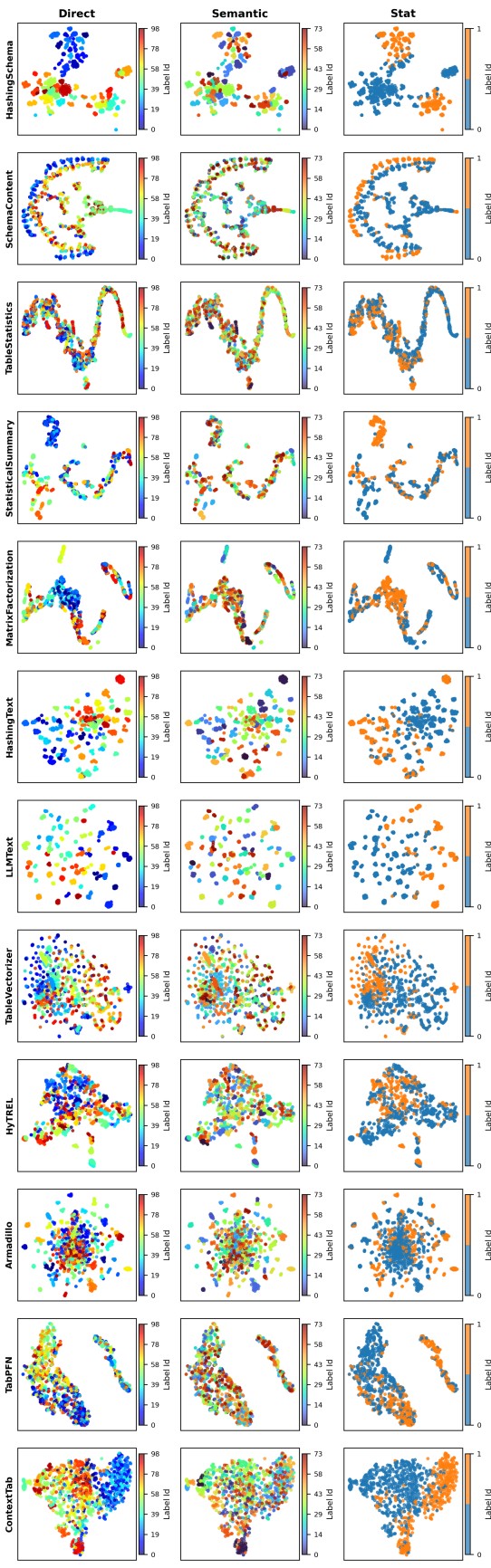

Figure 6: t-SNE projections of partition embeddings for real open-source data, colored by different labeling schemes (Direct, Semantic, Stat).

