# OpenReview forum: "Beyond Row-Level Prediction: A Unified Evaluation of Table Representation Methods and Recoverable Table-Level Geometry"
_TMLR — Under review for TMLR_

### Review · Reviewer_Sr5q · 2026-06-08

**Summary Of Contributions:**

This paper conducts a study of table representation methods and finds that the pre-training objectives (row-level predictions) used in tabular foundation models do not capture global table representations adequately. The authors design a benchmark and propose four practical desiderata to compare different methods. They generally find that non-neural baselines perform better than pre-trained encoders. They further evaluate an learned alternative to mean-pooling on top of a frozen encoder (ConTextTab) and find that the proposed alternative training outperforms mean-pooling. In addition to evaluation on synthetic and real-world tabular data, they also conduct an evaluation on identifying the parent table from table partitions.

**Audience:**

Yes

**Audience Explanation:**

The community of researchers interested in table representation learning will find the paper useful.

Having said that, I am not particularly convinced about the broader applicability of this work. Modern LLMs are very good at manipulating tabular data, e.g., by generating programs to analyze tables. The desiderate make sense in isolation but I would have liked to see how they are applicable in agentic pipelines, e.g., do they affect performance of RAG.

**Broader Impact Concerns:**

I don't have any ethical concerns about the paper.

**Claims And Evidence:**

Yes

**Claims Explanation:**

The paper is well-executed within its scope. The desiderata are explained and illustrated (Fig 1) nicely. The underlying metrics capture partial view consistency, ranking/grouping/retrieval, robustness to perturbations and efficiency of generating the representations. The coverage of tabular data grounded in real-world benchmarks and their synthetic augmentation is reasonable. A number of baselines (both neural and non-neural) are considered.

**Requested Changes:**

Add a study of how the proposed desiderata remain applicable in RAG and language agents operating on tabular data. Show that the existing conclusions remain valid in this setting.

---

> ### Author Response · Authors · 2026-06-19
> **Response to Reviewer Sr5q: Relation to Tabular RAG and Language-Agent Pipelines**
>
> We thank the reviewer for the positive assessment and for recognizing that the paper is well-executed within its scope. We also agree that tabular RAG and language-agent pipelines are important downstream motivations for reusable table representations.
>
> The reviewer raises an important question: **how do the proposed desiderata connect to agentic pipelines such as RAG?**
>
> In the revised manuscript, we clarify that our goal is not to evaluate a complete RAG or language-agent system, but rather to isolate a representation-level problem that such systems rely on: selecting relevant tables or table fragments before reasoning can take place. In a tabular RAG setting, an LLM would generate code, plan an analysis, or reason over evidence after the system has retrieved suitable tabular context from a corpus.
>
> This connection motivates our **parent-table retrieval proof of concept**. Given an incomplete table view, the embedding must retrieve the correct source table from a candidate corpus. This task directly reflects two of the benchmark’s central desiderata: **D1 partial-view consistency and D2 identity-sensitive discriminability**. The retrieval results are consistent with the benchmark: methods that preserve these properties also perform well in source recovery, while learned ConTextTab-R heads improve substantially over naive pooled ConTextTab.
>
> We revised the **Discussion (in section 8)**, highlighted in blue, to make this scope explicit. A complete tabular RAG or language-agent benchmark would additionally involve natural-language query interpretation, retrieval orchestration, re-ranking, prompting, program generation, and downstream reasoning. These are important system-level factors, but they would mix representation quality with many downstream design choices. Our contribution is therefore complementary: **we evaluate representation-level prerequisites for tabular retrieval and agentic table use, rather than end-to-end agent performance**.

---

### Review · Reviewer_TZP1 · 2026-06-10

**Summary Of Contributions:**

This paper studies reusable table-level representations, arguing that current tabular foundation models are typically evaluated through row-level prediction rather than through the quality of whole-table embeddings. The authors propose a unified evaluation framework based on four desiderata: consistency under partial views, discriminability across label granularities, robustness to benign perturbations, and efficiency. They compare a broad range of methods, including schema/statistical fingerprints, text-based serialization methods, specialized table encoders, and pooled representations from tabular foundation models such as TabPFN and ConTextTab.

In summary, the contributions are as follows

1. The benchmark design is clear and practically motivated. The four desiderata are well chosen and correspond to real use cases.
2. The finding that simple schema- and text-based methods outperform pooled tabular foundation model embeddings is valuable, especially because it highlights a potential gap between predictive performance and reusable representation quality.

**Audience:**

Yes

**Audience Explanation:**

The paper addresses an important and timely question in tabular machine learning: whether tabular foundation models produce reusable table-level representations, rather than only strong row-level predictions. So I believe at least some members of the TMLR audience would be interested in the findings of this paper.

**Claims And Evidence:**

No

**Claims Explanation:**

A primary concern is that some benchmark tasks may strongly favor schema and lexical overlap. Since column names, schema tokens, and serialized text are highly informative for parent-table retrieval and partial-view matching, the strong performance of lightweight methods may partly reflect benchmark design rather than a general advantage in table representation.

**Requested Changes:**

1. How do the results change if column names are removed, anonymised, or randomly renamed? This would help separate schema-matching ability from deeper table-content representation.

2. The learned-head case study is useful, but the training objective is closely aligned with the benchmark desiderata. This raises the question of whether the improvements reflect general-purpose table representation learning or optimisation toward the same proxy signals used for evaluation. Stronger held-out evaluations or more downstream tasks would help clarify this.

3. Can the learned ConTextTab heads generalise to tasks not directly related to the training objective, such as open-world table retrieval, semantic table search, or dataset recommendation?

---

> ### Author Response · Authors · 2026-06-19
> **Response to reviewer TZP1: Column-Name Anonymization, Learned-Head Scope, and Generalization**
>
> We thank the reviewer for the constructive assessment and for recognizing that the benchmark design is clear, practically motivated, and addresses an important question in tabular machine learning. We agree that the main evidence concern is whether the strong performance of schema/text baselines reflects useful table-level representation quality or is largely driven by schema and lexical overlap.
>
> The reviewer’s suggested **column-name anonymization test** is exactly the right diagnostic for this concern. We therefore added **a new ablation in the appendix A3** in which the evaluation is unchanged, but immediately before representation extraction we rename the columns of each partition positionally to col_1, ..., col_m. Cell values, dtypes, labels, coordinate sets, and partition geometry remain unchanged. This removes exact lexical column-name identity while preserving table content and structure.
>
> The results confirm the reviewer’s concern for schema-only methods: **HashingSchema drops substantially under anonymization**, with synthetic D1 Spearman decreasing from 0.58 to 0.29 and real-data D2 Avg decreasing from 0.69 to 0.50. Thus, we agree that part of HashingSchema’s original performance comes from column-name information.
>
> At the same time, **the ablation shows that the broader conclusion is not explained only by exact schema-name matching. HashingText remains largely stable**: on synthetic data, D1 changes from 0.54 to 0.55 and D2 Avg from 0.73 to 0.72; on real data, D1 changes from 0.40 to 0.35 and D2 Avg from 0.66 to 0.65. Statistical methods are also largely stable. We therefore revised the interpretation in the manuscript: **schema metadata is a realistic and useful signal in table discovery, but exact column-name matching is not the sole driver of the competitiveness of lightweight baselines.**
>
> Regarding **the learned-head case study**, we agree with the reviewer that the SCDM objective is aligned with the proposed desiderata, and we are aware that this limits how broadly the case-study results should be interpreted. We have therefore clarified in the revised manuscript that this experiment is **diagnostic rather than a claim of universal table representation learning**. Its purpose is to test whether useful table-level structure is present in frozen ConTextTab states but poorly composed by naive pooling.
>
> To avoid evaluating only on the training signal, we included two safeguards. First, as described in **Appendix A9**, the heads are selected and evaluated under separate training, validation, and final benchmark protocols, with different dataset roots and partition samplers. The final benchmark also includes synthetic evaluation although the heads are trained on real tables only. Second, **Section 7 evaluates the learned heads on parent-table retrieval**, which is not the SCDM training objective itself: the model must retrieve the correct source table from a candidate corpus given a partial table view.
>
> These results suggest that **the learned heads do generalize beyond the exact training configuration** to the paper’s source-recovery setting, but we do not overclaim this as evidence of open-world generalization. Open-world table retrieval, semantic table search, dataset recommendation, and full tabular RAG introduce additional system-level choices, such as query construction, corpus indexing, re-ranking, prompting, and downstream reasoning. **We therefore position them as important follow-up directions beyond the representation-centric scope of this work.**

---

### Review · Reviewer_66i6 · 2026-07-15

**Summary Of Contributions:**

The paper describes a benchmark for table level representations based on 4 characteristics the authors name as relevant:

1. consistency under partial views (measured as spearman rank correlation between the cosine similarity of the embeddings and the IoU of partitions)
2. discriminability across label granulatories (capturing both local and global structure,)measured via linear probes predicting labels from the embeddings, as well as triplet ranking, recall@K and clustering
3. robustness to benign perturbations (row and column permutation invariance, stabiliy under mild masking and noise)
4. throughput (tables per second)

Lable granulartiy ranges from "direct" (source label identity), "semantic" (tuples of generative factors for synthethic data,for real data tuples from benchmark source + dataset identifier), "semantic + difficulty" => semantic + an assigned label for the difficulty added to the tuple and "stat" (a coarse statistical signature (majority column type, single mixed label if no coarse type has clear majority )).

The benchmark explores a set of embedders (HashingSchema, SchemaContent, TableStatistics, StatisticalSummary, MatrixFactorization, HashingText, LLMText, TableVectorizer, HyTREL, Armadillo, TabPFN and ConTextTab) on a vaierty of datasets(four synthetic families with known generative factors - physics formulas, geometric shapes, temporal reasoning and structural causal model (SCM) priors - and the real open-source corpora CARTE, OpenML-CC18 and OpenML-CTR23) using mean pooling to yield the table embedding initially, then compares against learned aggregation methods (attention,gatedAttention,gatedMLP) and finds that very simple methods can compete with or outperform pretrained embedders.

The authors then discuss this as a a need for future work to take table level embeddings as an explicit design goal and not except simple aggregation methods to work to yield table embeddings from row or cell embedddings.

**Audience:**

Yes

**Audience Explanation:**

..with a caveat


If the message can be cleaned up, the leakage fixed, the motivation clarified, and the evaluation expanded across meaningful other aggregations and then remains the same, I could see the clear evidence of needing to approach table level representation of interest (especially if it can be shown that this was not happening before, see point 5 above)

**Claims And Evidence:**

No

**Claims Explanation:**

The paper is a bit all over the place and it is hard for me to see what the claims are meant to be taking it at its entirety (e.g. in the final section the paper disclaims that (emphasis mine)

>For that reason, we view the benchmark as *a principled first layer rather than a complete final answer. Its role is to expose a concrete failure mode*, provide a reproducible way to compare method families, and *reveal whether table-level geometry is recoverable from frozen predictive backbones*.

but then goes on to claim

> The paper is not only diagnostic.
where I get the point the authors are trying to make, but it would be clearer to simply clearly state the intended message of the paper instead of discussing what the paper is *not*.
)

For the claims made explcicitly (unified evaluation framework, showing lightweight schema dn text based methods often outperform naive pooled representations, gap recoverable via learning aggrtation heads, benchmark trends translate to meaningful tasks)

there are a few issues that limit me from giving a yes

1. the metrics chosen align well with the winning method (e.g. an aggregated hash based fingerprint *is* a form of IoU measurement)
2. No other "simple" or data agnostic aggregation methods used (max pool? the set NN and GNN community has many different pooling methods )
3. there's a data leakage problem in the code which might affect resutls (no group stratification used, just label stratification which could falsify labels)
4. Missing related work and prior art to contextualize (see requested changes)
5. Missing motivations: did anyone *expect* meanpooling to work directly (already word to sentence and sentence to document were known to not work that well after Word2Vec )? Why is the case study task a relevant task? Were applications in table based ML not using the methods discussed in the implication so far?

**Requested Changes:**

1. Please add/diuscuss the missing citations below or justify why not, and sitaute your benchmark witin them
2. Compare against maxpool, and maybe other existing more sophisticated but not learned pooling methods
3. fix the leakage bug (or clarify that it was not an issue)
4. small list of typos
   1. Section A.2.3 sends the reader to "Figures 2 and 3 in the Appendix A.6" for the per-label D2 breakdown, but that breakdown is in Tables 11–19; Figures 2 and 3 are the main-text summary bar charts (pp. 6–7).
   2. The captions of Tables 21 and 22 read "(mean ± 95% CI over seeds)", dropping the seed count "10" that every other results caption gives (cf. Tables 2–5).
   3. "ConTextTab-R" (Tables 5, 20) is never expanded, and the same learned-head models are also written "ConTextTab + Attention (α=…)" (Tables 21–22) and as bare "Attention/GatedAttention/GatedMLP heads" (Sec. 6) -three notations for one object; please pick one and define it.
   4. TR-CH is called "conditional-hard" in Sec. 5.1 but "cluster-hard" where it is defined (A.2.3).
   5. Stray space before punctuation in the D3 desideratum ("…small additive noise ." in Sec. 3).

#### Missing citations to add or justify omitting

Some concurrent work which might not have been visible to the authors but seems reelvant:

- **TEmBed -Towards Universal Tabular Embeddings** (Vogel, Srinivas, D'Souza, Shirai, Hassanzadeh & Samulowitz, 2026) -a benchmark evaluating tabular embeddings across cell/row/column/*table* levels, concluding the best model depends on task and level; the closest concurrent competitor. <https://arxiv.org/abs/2604.21696>
- **TabBench / TabEmbed** (Qiang et al., 2026, <https://arxiv.org/abs/2605.04962>) and **TRL-Bench** (Pang et al., 2026, <https://arxiv.org/abs/2606.09323>) -two further concurrent multi-granularity tabular-embedding benchmarks.

Foundational table encoders the benchmark positions itself against but does not cite:

- **TaBERT** (Yin, Neubig, Yih & Riedel, ACL 2020) -the canonical joint text–table transformer and a root of the "pretrained table encoder" lineage the paper argues against. <https://aclanthology.org/2020.acl-main.745/>
- **TABBIE** (Iida, Thai, Manjunatha & Iyyer, NAACL 2021) pretrains purely on tables and explicitly emits cell-, row-, and table-level representations; the most directly comparable prior art to "table-level representation via pooling of cell/row states." <https://aclanthology.org/2021.naacl-main.270/>
- **TURL** (Deng, Sun, Lees, Wu & Yu, PVLDB 2020) -pretrain/fine-tune framework for relational-table representations shipped with a table-understanding benchmark; it already frames "evaluate table representations across tasks," which the paper claims is under-specified. <https://arxiv.org/abs/2006.14806>

The downstream tasks the paper says its trends translate to -table discovery/retrieval -where whole-table representations are already computed and ranked:

- **Table Union Search on Open Data** (Nargesian, Zhu, Pu & Miller, PVLDB 2018) the parent-table retrieval proof-of-concept seems to be a special case of it. <http://www.vldb.org/pvldb/vol11/p813-nargesian.pdf>
- **Starmie** (Fan, Wang, Li, Zhang & Miller, PVLDB 2023) -contrastive column embeddings evaluated  by cosine-similarity table-union retrieval. <https://arxiv.org/abs/2210.01922>
- **TabSketchFM** (Khatiwada et al., 2024) -learns whole-table embeddings for data-lake union/join/subset discovery. <https://arxiv.org/abs/2407.01619>

Optional, but I think relevant and egnagement would strengthen the work:

- **"Something's Fishy in the Data Lake: A Critical Re-evaluation of Table Union Search Benchmarks"** (Boutaleb, Amann, Naacke & Angarita, 2025) -shows plain lexical/SBERT baselines match specialized table encoders on union-search benchmarks because the ground truth is dominated by column-name overlap; also reports "simple methods win" result and argues it is partly a benchmark artifact=>  most important omission see issue 1 above. <https://arxiv.org/abs/2505.21329>
- **A Metric Learning Reality Check** (Musgrave, Belongie & Lim, ECCV 2020) -documents that embedding/retrieval benchmarks are routinely inflated by test-set leakage, weak baselines, and single-number reporting, and prescribes fairer protocols =>directly relevant to the leakage bug (issue #3) and to reading method rankings off overlapping intervals with no significance test (less of an issue here I think because the gaps seem clear, but still a point about rigor). <https://arxiv.org/abs/2003.08505>